# Flick: Empowering Federated Learning with Commonsense Knowledge

**Ran Zhu[1], Mingkun Yang[1], Shiqiang Wang[2], Jie Yang[1], Qing Wang[1]**

[1]Delft University of Technology, [2]IBM Research
{r.zhu-1, m.yang-3, j.yang-3, qing.wang}@tudelft.nl
shiqiang.wang@ieee.org

## Abstract

Federated Learning (FL) has emerged as a privacy-preserving framework for training models on data generated at the edge. However, the heterogeneity of data silos (e.g., label skew and domain shift) often leads to inconsistent learning objectives and suboptimal model performance. Inspired by the data-driven approach, we propose Flick, a novel data generation framework for heterogeneous **F**ederated **L**earning w**i**th **C**ommonsense **K**nowledge from Large Language Models (LLMs). In Flick, the client performs the local data summary to capture client-specific knowledge in textual form. The central server then distills task-relevant, high-quality knowledge from the out-of-the-box LLM – guided by cross-client-specific insights – to generate informative text prompts. These prompts direct a generative model in producing synthetic data, enabling global model fine-tuning and local data compensation. This process gradually aligns the label and feature distributions across clients. Extensive results on three datasets demonstrate that Flick improves the global model accuracy by up to 11.43%, and accelerates convergence by up to 12.9×, validating its effectiveness in addressing data heterogeneity. The code can be found at `https://github.com/Ran-ZHU/Flick`.

## 1 Introduction

With the proliferation of mobile devices equipped with sensing capabilities, edge-generated data has unlocked new opportunities for cyber-physical services. Meanwhile, advances in computing and networking at the edge have driven the adoption of edge computing, where deploying deep learning models near data sources enables agile and efficient AI applications within distributed systems [1–3]. In this context, Federated Learning (FL) has emerged as a key paradigm that enables collaborative model training on decentralized data while preserving data privacy [4–6]. FL proceeds in communication rounds, where participating devices (clients) perform several steps of Stochastic Gradient Descent (SGD) on local data and send the updated models to a central server for aggregation into a global model. A commonly used algorithm is FedAvg [5], which alternates between local updates on clients and global parameter aggregation among clients by a central server. However, the performance of the global model is often hindered by the non-identically and/or non-independently distributed (non-IID) nature of data silos across clients. Specifically, when data distributions are heterogeneous – due to class imbalance or domain shifts (e.g, sketch vs. photo) – clients may learn inconsistent objectives. This leads to slower convergence and reduced generalization in the resulting global model (Figure 1(left)).

To solve this problem, previous works proposed solutions such as 1) modifying loss functions for local models [7–10], 2) re-weighting central aggregation for the global model [11–13], or 3) using adaptive hyper-parameters for local training [14–16]. However, since the local objective is only computable on clients, those *model-driven* methods that align the global optimum with local (surrogate) objectives

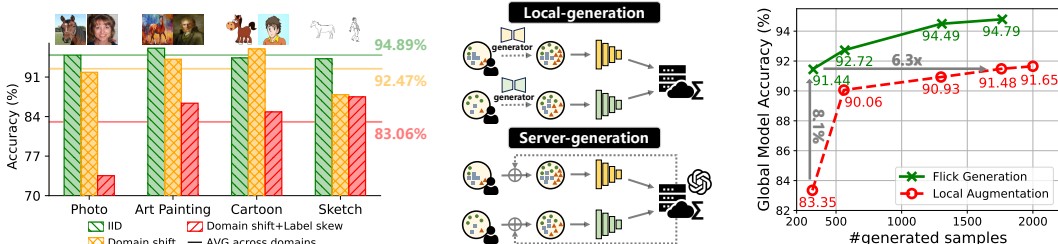

Figure 1: **Problem (left figure):** Heterogeneous data across clients–specifically label skew and domain shift–leads to cross-domain variance and inferior overall accuracy, rendering the degraded global model in FL. **Solution (middle figure):** In *data-driven* approaches, most existing frameworks compensate for heterogeneous data by generating samples locally. However, a server-based generation framework offers the potential to run computationally intensive image generation models, thus providing high-quality synthetic samples. **Flick's Gain (right figure):** Flick improves the test accuracy by a large margin while requiring fewer generated samples.

cannot fundamentally eliminate the model bias caused by heterogeneous local data. A few recent works aiming at addressing this issue are based on *data-driven* methods by utilizing generative models – either on the server or client side – to produce pseudo samples or synthetic data for fine-tuning the global model or enriching local datasets, which requires sensitive information shared between clients and the server such as local data distribution, a small portion of local data or a large number of data features [17–20]. These approaches raise severe privacy concerns. On the other hand, many frameworks [18, 21–23] perform generative models directly on the client side to avoid transmitting privacy information to the server. While this allows for efficient data augmentation tailored to locally observed data, only client-specific augmentation remains insufficient for addressing domain shift challenges. Also, the effectiveness and efficiency of local generation are inherently constrained by the limited scope of knowledge and computational resources of clients. We ask the following question:

*Can we design a generative framework utilizing limited low-sensitivity cross-client knowledge and task-relevant commonsense of LLMs to quickly promote FL's performance under data heterogeneity?*

In this paper, we propose **Flick**, a novel server-side data generation framework to mitigate data heterogeneity, with a particular focus on **label skew** and **domain shift**. Flick, in a nutshell, provides a novel design to distill client-specific knowledge while leveraging Large Language Models (LLMs) – as an extra informative source – to instill task-relevant commonsense knowledge into data generation. The synthetic data is then used to refine both local data silos and the global model. Specifically, at the beginning of each communication round, the participating client selectively captions the local samples using a pre-trained image-to-text model [24] and reports the local summaries (in the form of token sequences) to the server. This process extracts and converts salient yet sensitive local information into a relatively low-sensitive textual format. The central server in Flick then employs a carefully designed prompt template to instruct an out-of-the-box LLM in analyzing local information (referred to as cross-client-specific knowledge) while incorporating the LLM's inherent commonsense knowledge to produce informative text prompts. These text prompts are subsequently fed into an image-generation model, benefiting from the powerful computational resources to run a computationally intensive model on the server side, and thus can generate high-quality synthetic samples. These data points serve two key purposes: 1) the server sends the generated samples back to the clients to compensate for heterogeneous data silos, and 2) the server fine-tunes the aggregated global model on the generated samples. These designs distinguish Flick from existing data augmentation methods. For the gain (Figure 1(right)), Flick improves the global model accuracy by up to 8.1% compared to local augmentation with the same amount of generated samples, and the global model in Flick achieves comparable accuracy to local augmentation while the latter requires 6.3× generated samples. We attribute this performance gap to the effective distillation of task-relevant commonsense knowledge from LLMs, which is further facilitated by cross-client-specific insights. In summary, we make the following key contributions:

- We design Flick, a framework to enhance FL's performance on non-IID data by blending local and server-generated samples. It integrates low-sensitive cross-client-specific knowledge and LLMs' commonsense knowledge, enabling clients to progressively acquire unbiased data in both label and domain aspects.

- We design a local summary method to encode client-specific knowledge into token sequences to facilitate task-specific data generation at the server. The generated samples serve two purposes – local data compensation and global model fine-tuning – to enhance FL's overall performance.

- Extensive evaluations on three datasets demonstrate Flick significantly enhances FL's performance, achieving up to 11.43% higher model accuracy and $12.9\times$ faster convergence. Moreover, our study highlights Flick as a flexible pipeline adaptable to various LLMs and image-generation models.

## 2 Related Work

Existing works to solve heterogeneous FL can broadly be classified as model-driven and data-driven.

**Model-driven Heterogeneous FL.** Model-driven methods seek to mitigate the diverse model updates caused by inconsistent local objectives through modifications to the loss function, robust aggregation methods, or adaptive hyper-parameter settings. For the skewed label distribution, some works [7–9] introduce penalty terms into the local loss function. Alternatively, FedNova [12] re-weights the local updates during central aggregation, while FedOpt [13] and FedAvgM [11] take aggregation as an optimization problem and apply various optimizers to average the updates. In terms of domain shift, FedHEAL [25] selectively aggregates weight parameters based on their importance to performance improvement, and FedBN [26] excludes the parameters of batch normalization layers from aggregation. Futhermore, concurrent frameworks also adapt hyper-parameters across communication rounds to solve the data heterogeneity issue, such as learning rate and weight decay for SGD solvers [16] or the proportion of participating clients [15].

**Data-driven Heterogeneous FL.** In data-driven paradigms, auxiliary data/pseudo samples are synthesized via generative models deployed on either the client or server side, which are leveraged to refine the global model or augment local training distribution. Based on the entity generating data, works can be categorized into local generative and server generative approaches. In local generative methods, clients typically maintain a local generator that is periodically updated. For instance, the conditional Generative Adversarial Network (GAN) in FAug [17] and the Conditional AutoEncoder (CVAE) in FedDA [18]. The Gen-FedSD [22] and ReGL [23] generate synthetic data more directly by feeding a local Stable Diffusion (SD) model with textual prompts from predefined templates. Alternatively, FRAug [21] performs augmentation in the feature embedding space rather than in the input space. In server generative methods, such as FedFTG [19] and DynaFed [27], pseudo-samples are generated based on knowledge distilled from the global model. FGL [20] shares a similar motivation with Flick, as both use client-side information to guide server-side data generation. However, FGL requires clients to caption all local images, incurring substantial overhead and exposing sensitive information such as local data distributions. It also relies on large volumes of synthetic data for global model fine-tuning, increasing computational cost. In contrast, Flick minimizes the client-side effort during the local summary phase, while achieving efficient data generation by integrating cross-client-specific insights with commonsense knowledge distilled from out-of-the-box LLMs.

## 3 Flick Design

In FL, the server searches for the optimal global model $\mathcal{W}^*$ by aggregating local updates from a set of clients $\mathcal{J}$, iterating over $N \in \mathbb{N}$ communication rounds. In round $n \in [N]$ ($[N] = \{1, \cdots, N\}$), a subset of clients $\mathcal{J}^n \subseteq \mathcal{J}$ perform SGD on their local data $\{\mathcal{D}_j\}_{j \in \mathcal{J}}$ in parallel where $\mathcal{D}_j = \{(x_j^k, y_j^k) \in \mathbb{R}^d \times \mathbb{N}^C | k \in [|\mathcal{D}_j|]\}$, with $d$, $C$ (number of classes) representing the dimensions of input and output space. The global objective function $\mathcal{L}(\mathcal{W})$ is the weighted sum of local objectives:

$$\mathcal{W}^* = \arg\min_{\mathcal{W}} \mathcal{L}(\mathcal{W}) = \arg\min_{\mathcal{W}} \sum_{j \in \mathcal{J}} p_j \mathcal{L}_{\mathcal{D}_j}(\mathcal{W}), \tag{1}$$

where the weights for local objectives $\{\mathcal{L}_{\mathcal{D}_j}(\mathcal{W})\}_{j \in \mathcal{J}}$ satisfy $\sum_{j \in \mathcal{J}} p_j = 1$; the fraction of local samples $p_j = |\mathcal{D}_j| / \sum_{j \in \mathcal{J}} |\mathcal{D}_j|$. Specifically, the local objective of $j$-th client is the empirical loss over the local data, that is, $\mathcal{L}_{\mathcal{D}_j}(\mathcal{W}) = \sum_{(x,y) \in \mathcal{D}_j} l(\mathcal{W}; x, y) / |\mathcal{D}_j|$ where $l$ is the loss function.

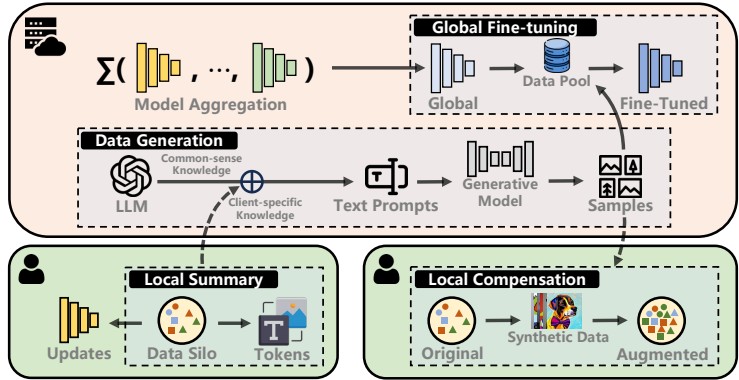

Figure 2: Flick overview: it bootstraps participating clients to summarize their local data, distilling client-specific knowledge into low-sensitivity tokens. The central server uses an out-of-the-box LLM to analyze local summaries and instill inherent commonsense knowledge into generated text prompts. A text-to-image model then produces synthetic samples, which are used in two ways: to compensate for the local datasets of clients and to fine-tune the aggregated global model.

In the context of data heterogeneity, the local data distribution $\mathcal{D}_j \sim \mathcal{P}_j(x, y)$ differs between clients, meaning $\mathcal{P}_{j_1}(x, y) \neq \mathcal{P}_{j_2}(x, y), \forall j_1, j_2 \in \mathcal{J}, j_1 \neq j_2$. Following the taxonomy of non-IID data in [28], we rewrite $\mathcal{P}_j(x, y)$ as $\mathcal{P}_j(x|y)\mathcal{P}_j(y)$. In this paper, we focus on two types of data heterogeneity: label skew and domain shift, where the marginal distribution $\mathcal{P}_j(y)$ and the conditional distribution $\mathcal{P}_j(x|y)$ may vary across clients, respectively. To address this problem, we propose Flick, a *data-driven* approach where the central server generates new data points $\tilde{\mathcal{D}}_j^n$ for each participating client $j \in \mathcal{J}^n$. In this way, the local data is progressively updated over communication rounds, such that $\mathcal{D}_j^n = \mathcal{D}_j^{n-1} \cup \tilde{\mathcal{D}}_j^n$. For the stale client $j \notin \mathcal{J}^n$, the local dataset remains unchanged, i.e., $\mathcal{D}_j^n = \mathcal{D}_j^{n-1}$. The data across clients progressively approaches an IID condition, where $\mathcal{P}_{j_1}(x|y) \simeq \mathcal{P}_{j_2}(x|y)$ and $\mathcal{P}_{j_1}(y) \simeq \mathcal{P}_{j_2}(y)$ with $\forall j_1, j_2 \in \mathcal{J}, j_1 \neq j_2$. All notations used throughout this paper are summarized in the Appendix A.

## 3.1 Flick Overview

The overall framework of Flick is illustrated in Figure 2. Flick introduces two additional phases into the FL workflow: local summary and data generation. Clients first summarize their local data by selectively captioning samples. The server then generates new data points by analyzing the collected captions using an LLM guided by a designed prompt, fusing client-specific insights with commonsense knowledge embedded in the LLM. Generated samples are stored in a server-maintained data pool, which supports both global model fine-tuning and local data compensation. Flick is designed to ensure effective data generation and efficient utilization of synthetic data, aiming to enhance global performance and convergence. However, designing Flick is non-trivial: 1) the server requires clients to report information about local data, raising potential privacy concerns; 2) data generation has to balance the sample quality and sample quantity within limited budgets (e.g., maximum generation cost, generation latency), necessitating careful prompt design and generative database management; 3) two types of generated sample usage are complementary and equally important for improving global model performance: progressively providing clients with additional samples to approach the IID condition yields long-term benefits while fine-tuning the global model provides a more straightforward yet slight improvement. Detailed designs are presented in the following sections.

## 3.2 Local Summary Phase

In round $n$, clients $\mathcal{J}^n$ initialize their models with the latest global weights $\mathcal{W}_j^{(n,0)} = \mathcal{W}^{n-1}$, and perform $\tau_j \in \mathbb{Z}^+$ steps of SGD on the local data $\mathcal{D}_j^{n-1}$: $\mathcal{W}_j^{(n,\tau_j)} = \mathcal{W}^{n-1} - \eta \sum_{k=0}^{\tau_j-1} \nabla \mathcal{L}_{\mathcal{B}_j^{(n,k)}}(\mathcal{W}_j^{(n,k)})$, where $\nabla \mathcal{L}_{\mathcal{B}_j^{(n,k)}}(\mathcal{W}_j^{(n,k)})$ is stochastic gradient over a mini-batch $\mathcal{B}_j^{(n,k)} \subseteq \mathcal{D}_j^{n-1}$ at step $k \in [\tau_j]$. $\eta$

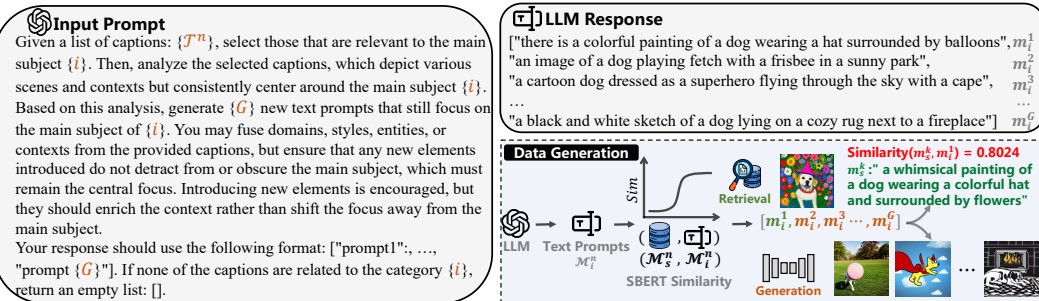

Figure 3: Illustration of the data generation phase: by feeding the designed prompt containing cross-client-specific knowledge $\mathcal{T}^n$, the server obtains the text prompts for data generation in the $i$-th class (e.g., dog) from the LLM output. To reduce generation cost, Flick first retrieves historical samples from the server-held data pool based on the Sentence-BERT (SBERT) embeddings [36] similarity between the LLM-generated and historical text prompts. For remaining text prompts that fail to find matches in the historical data, the server uses a generative image model to produce synthetic samples.

is the learning rate. The post-training summary phase extracts client-specific knowledge from local data, providing the server with references for data generation. To achieve this, each participating client selects representative samples and uses an image-to-text model as a caption generator to transform samples into low-sensitivity textual information, which is then offloaded to the server.

**Sample Selection.** To minimize the privacy risk and local captioning overheads, the representative samples $\hat{\mathcal{D}}_j^{n-1} \subseteq \mathcal{D}_j^{n-1}$ of the $j$-th client for captioning should be limited in size, while still ensuring coverage of all classes held by the client. In this way, we propose a loss-based sample selection strategy where samples are chosen based on their informativeness for the updated model $\mathcal{W}_j^{(n,\tau_j)}$. Specifically, client $j$ calculates the class-wise loss of the local model across the holding classes $\mathcal{C}_j^{n-1}$:

$$ l_{j,c}^n = \frac{1}{|\mathcal{D}_{j,c}^{n-1}|} \sum_{(x,y) \in \mathcal{D}_{j,c}^{n-1}} l(\mathcal{W}_j^{(n,\tau_j)}; x, y), \tag{2} $$

where $\mathcal{D}_{j,c}^{n-1}$ refers to the set of local samples in the class $c$, and $\mathcal{D}_j^{n-1} = \bigcup_{c \in \mathcal{C}_j^{n-1}} \mathcal{D}_{j,c}^{n-1}$. We define the average class-wise loss as $\bar{l}_j^n = \sum_{c \in \mathcal{C}_j^{n-1}} l_{j,c}^n / |\mathcal{C}_j^{n-1}|$.

The sample selection follows the criteria: for each class $c \in \mathcal{C}_j^{n-1}$, if the class-wise loss $l_{j,c}^n \leq \bar{l}_j^n$, we select the sample from $\mathcal{D}_{j,c}^{n-1}$ with the largest loss in that class. Conversely, if $l_{j,c}^n > \bar{l}_j^n$, we select the sample with the smallest loss. The rationale behind this is that, for classes where the updated model performs above average, we choose the sample with the largest loss, as it has a significant influence on model training [29]. For classes where the model has inferior performance, selecting the sample with the smallest loss ensures that the representative samples are less likely to be noisy or distorted.

**Sample Captioning.** Each clients extracts a set of representative samples $\hat{\mathcal{D}}_j^{n-1}$ for captioning. As the text generation task, the learning-based image captioning provides natural language descriptions for visual content using the encoder-decoder model structure [30–32]. Among existing methods, we employ VLP model [33, 24, 34] pre-trained on large-scale image-text pairs for local captioning. Each client feeds images $\hat{\mathcal{D}}_j^{n-1}$ into the VLP model for inference, generating a set of output token sequences $\tilde{\mathcal{T}}_j^n$, which are captions for corresponding representative samples.

**Privacy Consideration.** Compared to frameworks where clients offload raw local data or latent features to the server [17, 18], Flick has a privacy-conscious design for the local summary, adhering to the data minimization and anonymization principles defined in work [35]. Specifically, it offers a minimally invasive method by selectively captioning a small subset of representative local samples into token sequences, providing the server with essential yet low-sensitive information for data generation. We also conduct the privacy assessment of the local summary design in Appendix E.

### 3.3 Data Generation Phase

In round $n$, the central server collects locally updated models $\{\mathcal{W}_j^{(n,\tau_j)}\}_{j\in\mathcal{J}^n}$ and token sequences of local representatives $\{\tilde{\mathcal{T}}_j^n\}_{j\in\mathcal{J}^n}$. The server aggregates local updates for a global model $\bar{\mathcal{W}}^n = \frac{1}{\sum_{j\in\mathcal{J}^n}|\mathcal{D}_j|}\sum_{j\in\mathcal{J}^n}|\mathcal{D}_j|\mathcal{W}_j^{(n,\tau_j)}$. In parallel, the server decodes the token sequences $\tilde{\mathcal{T}}_j^n$ into the text captions $\mathcal{T}_j^n$ and pools them into $\mathcal{T}^n = \bigcup_{j\in\mathcal{J}^n}\mathcal{T}_j^n$. Based on the analysis of $\mathcal{T}^n$ supported by LLMs, the server provides clients with new data points $\{\tilde{\mathcal{D}}_j^n\}_{j\in\mathcal{J}^n}$, where $\tilde{\mathcal{D}}_j^n = \{(x_j^k, y_j^k) \in \mathbb{R}^d \times \mathbb{N}^C | k \in [\|\tilde{\mathcal{D}}_j^n\|]\}$, to compensate for local data distribution by blending each local dataset $\mathcal{D}_j^{n-1}$. Simultaneously, the server fine-tunes the aggregated global model with generated samples to update the model weights from $\bar{\mathcal{W}}^n$ to $\mathcal{W}^n$.

**Generative Data Pool.** The server in Flick maintains a generative dataset $\mathcal{G}_s = \{(x_s^k, y_s^k, m_s^k)|k \in [\|\mathcal{G}_s\|]\}$ to pool the pairs of data points $(x_s, y_s)$ along with corresponding text prompts $m_s$ used to generate each data point. Specifically, by feeding text prompts into a generative model $f^{\mathcal{W}_G}$, for instance, a latent diffusion model (LDM) [37], the server generates synthetic sample $x = f^{\mathcal{W}_G}(m)$ in the class $y$. The server-held dataset $\mathcal{G}_s$ is tightly coupled with the data generation process by using it in three ways: as a validation dataset for evaluating local updates on task-specific classes, as a database for retrieving historical samples, and as a fine-tuning dataset for global model enhancement. Compared to previous frameworks [38–40] constructing the public/auxiliary dataset residing in the server, $\mathcal{G}_s$ differs in two aspects: 1) the public dataset typically requires either web-sourced data or soliciting data from paid anonymous workers, whereas $\mathcal{G}_s$ in Flick is entirely self-contained and generated from scratch; 2) server-side datasets in previous work are usually static while $\mathcal{G}_s$ is dynamic and evolves over the communication rounds by updating with new samples.

**Synthetic Data Generation** To generate the task-required samples, the server first determines class-level data requirements for each client based on validation accuracy of local models on the current server-side dataset $\mathcal{G}_s^{n-1}$, as shown in Algorithm 1 (**line 1-2**). We use matrix $\mathbf{D}^n \in \mathbb{N}^{C\times|\mathcal{J}^n|}$ to record decisions in $n$-th round, where the entry $\mathbf{D}_{i,j} \in \{0,1\}$ in the $i$-th row and $j$-th column indicates whether supplementing client $j$ with samples of class $i$. The value of $\mathbf{D}_{i,j}$ depends on the performance of local model: $\mathbf{D}_{i,j} = 1$ when the validation accuracy of $\mathcal{W}_j^{(n,\tau_j)}$ on class $i$ is below the predefined threshold $T_v$, suggesting that client $j$ would benefit from additional samples of class $i$ to improve local training, and $\mathbf{D}_{i,j} = 0$ otherwise.

The server then generates samples for classes $\mathcal{I}^n = \{i|i \in [C], \sum_{j\in\mathcal{J}^n}\mathbf{D}_{i,j} > 0\}$. The generation procedure follows three steps: 1) Given a generation budget $G$ (equal for each class), an out-of-the-box LLM generates text prompts $\mathcal{M}_i^n = \{m_i^k|k \in [G]\}$ following the designed prompt shown in Figure 3. The prompt template instructs LLM in extracting information from the provided cross-client-specific knowledge $\mathcal{T}^n$ while instilling inherent commonsense knowledge to produce informative text prompts for the $i$-th class. 2) To reduce generation overheads, the server retrieves historical samples from the data pool $\mathcal{G}_s^{n-1}$ by comparing the similarity between generated

---

**Algorithm 1:** Data generation and usage.

> **input** : Captions $\mathcal{T}^n$; Local updates $\{\mathcal{W}_j\}_{j\in\mathcal{J}^n}$; Sever-held dataset $\mathcal{G}_s^{n-1}$; Thresholds $T_v, T_s$; Generative model $f^{\mathcal{W}_G}$; Budget $G$.
> **output**: Fine-tuned global model weights $\mathcal{W}^n$; Compensated local dataset $\{\mathcal{D}_j\}_{j\in\mathcal{J}^n}$.

**Server Executes Data Generation:**
/* obtain decision matrix */
1   $\mathrm{Val}_{C\times|\mathcal{J}^n|} \leftarrow$ validate $\{\mathcal{W}_j\}_{j\in\mathcal{J}^n}$ on $\mathcal{G}_s^{n-1}$
2   $\mathbf{D}^n_{C\times|\mathcal{J}^n|} \leftarrow \mathbb{1}(\mathrm{Val} \leq T_v)$
3   $\mathcal{M}_s^n \leftarrow$ extract historical text prompts in $\mathcal{G}_s^{n-1}$
4   **for** $i = 1, 2, \cdots, C$ **do**
5     **if** $\sum_{j\in\mathcal{J}^n}\mathbf{D}_{i,j} > 0$ **then**
     /* obtain text prompts */
6      $\mathcal{M}_i^n \leftarrow$ feed LLM prompt $(\mathcal{T}^n, G, i)$
7      $\mathrm{Sim} \leftarrow$ pair-wise similarity $(\mathcal{M}_i^n, \mathcal{M}_s^n)$
     /* retrieval-based samples */
8      $(\mathcal{M'}_i^n, \mathcal{M'}_s^n) \leftarrow \mathbb{1}(\mathrm{Sim} \geq T_s)$
9      $\mathcal{D}_{\mathrm{retr}}^{(n,i)} \leftarrow$ samples from $\mathcal{G}_s^{n-1}$ by $\mathcal{M'}_s^n$
     /* generation-based samples */
10     $\mathcal{D}_{\mathrm{gen}}^{(n,i)} \leftarrow$ feed $f^{\mathcal{W}_G}$ text prompts $\mathcal{M}_i^n \setminus \mathcal{M'}_i^n$
11     $\mathcal{D}_i^n \leftarrow \mathcal{D}_{\mathrm{retr}}^{(n,i)} \cup \mathcal{D}_{\mathrm{gen}}^{(n,i)}$

/* local data compensation */
12   **for** $j \in \mathcal{J}^n$ **do**
13     $\tilde{\mathcal{D}}_j^n \leftarrow \bigcup_{i\in[C]} \mathcal{D}_i^n \times \mathbb{1}(\mathbf{D}_{i,j} = 1)$
14     $\mathcal{D}_j^n \leftarrow$ send $\tilde{\mathcal{D}}_j^n$ to client $j$ executing $\mathcal{D}_j^{n-1} \cup \tilde{\mathcal{D}}_j^n$
/* global model fine-tuning */
15   $\mathcal{G}_s^n \leftarrow$ update server-held dataset by $\bigcup_{i\in[C]} \mathcal{D}_i^n$
16   $\mathcal{W}^n \leftarrow$ fine-tune global model by $\mathcal{G}_s^n$

---

text prompts $\mathcal{M}_i^n$ and pooled text prompts $\mathcal{M}_s^n = \{m_s^k | m_s^k \in \mathcal{G}_s^{n-1}\}$. We calculate the cosine similarity of Sentence-BERT (SBERT) embeddings [36] for each pair of text prompts from $\mathcal{M}_i^n$ and $\mathcal{M}_s^n$. For the text prompts $\mathcal{M}'^n_i \subseteq \mathcal{M}_i^n$ where the highest pairwise similarity with historical text prompts $\mathcal{M}'^n_s \subseteq \mathcal{M}_s^n$ exceeds a predefined threshold $T_s$, the server includes the corresponding data points $\mathcal{D}_{\text{retr}}^{(n,i)} = \{(x_s^k, y_s^k) | (x_s^k, y_s^k, m_s^k) \in \mathcal{G}_s^{n-1}, m_s^k \in \mathcal{M}'^n_i\}$, thereby avoiding the need to generate new samples. 3) For the remaining text prompts $\mathcal{M}_i^n \setminus \mathcal{M}'^n_i$, the server employs an image generator $f^{\mathcal{W}_G}(\cdot)$ to synthesize new data points $\mathcal{D}_{\text{gen}}^{(n,i)} = \{(f^{\mathcal{W}_G}(m_i^k), y_i^k) | m_i^k \in \mathcal{M}_i^n \setminus \mathcal{M}'^n_i\}$. In this way, the server provides each class $i \in \mathcal{I}^n$ with samples obtained by either retrieval or generation: $\mathcal{D}_i^n = \mathcal{D}_{\text{retr}}^{(n,i)} \cup \mathcal{D}_{\text{gen}}^{(n,i)}$. Algorithm 1 **(line 5-11)** describes the overall data generation process. More data generation details, including synthetic samples, are provided in Appendix G.

### 3.4 Generated Data Usage

The server uses the generated samples in two ways: for global model fine-tuning and for local data compensation, as described in Algorithm 1 **(line 12-16)**. The server sends the corresponding generated samples to $j$-th client: $\tilde{\mathcal{D}}_j^n = \bigcup_{i \in \mathcal{I}^n} \mathcal{D}_i^n \cdot \mathbb{1}(\mathbf{D}_{i,j} = 1)$, where $\mathbb{1}(\cdot)$ is the indicator function. Each client then compensates for its local data distribution by updating $\mathcal{D}_j^n = \mathcal{D}_j^{n-1} \cup \tilde{\mathcal{D}}_j^n, \forall j \in \mathcal{J}^n$. In parallel, Flick updates the server-held data pool $\mathcal{G}_s^{n-1}$ by replacing the stale samples with latest generated samples $\bigcup_{i \in \mathcal{I}^n} \mathcal{D}_i^n$, keeping the dataset size constant. The server then uses the updated dataset $\mathcal{G}^n s$ to fine-tune the aggregated global model, updating the weights from $\bar{\mathcal{W}}^n$ to $\mathcal{W}^n$.

## 4 Experiments

### 4.1 Experimental Setup

**Datasets and Models.** We extensively evaluate Flick on the multi-domain image classification task, where data of different domains exhibit heterogeneous appearances but share the same labels. We use three datasets: **(1) PACS** [41], consisting of 9,991 images in 7 classes across the following four domains: *Photo, Art Painting, Cartoon, and Sketch*; **(2) Office-Caltech** [42], containing 10 overlapping classes between the Office dataset [43] and Caltech256 dataset [44], with data from four domains: *Amazon, Caltech, DSLR, and Webcam*; **(3) DomainNet** [45], a large-scale benchmark covering six domains: Clipart, Infograph, Painting, Quickdraw, Real, and Sketch, each originally containing 345 object classes. Following prior work [20, 26, 46], we construct a subset of DomainNet by selecting the top 10 most common classes for our experiments.

In our experiments, we simulate federated settings with heterogeneous data distributions. For the PACS dataset, we use 20 clients, and for Office-Caltech, we adopt 8 clients. Data from each domain is partitioned into 5 (PACS) or 2 (Office-Caltech) subsets using a Dirichlet distribution with concentration parameter $\alpha = 0.1$ and $\alpha = 0.05$, respectively. To evaluate the scalability of Flick in large-scale federated settings, we further conduct experiments on the DomainNet dataset with 100 clients, where 20% are randomly selected to participate in each communication round. Each domain is split into 15 or 17 subsets using a Dirichlet distribution with $\alpha = 0.1$. Across all three datasets, each client receives data from a single domain with a skewed label distribution, effectively simulating real-world scenarios characterized by both domain shift and label skew. Following prior works [21, 25, 47], we employ ResNet-18 as the shared model architecture for PACS and DomainNet, and ResNet-10 [48] for Office-Caltech across all compared methods.

**Baselines and Metric.** Our evaluation is based on four baselines: the vanilla FedAvg [5] and three model-driven methods– FedProx [9], FedDyn [49], and FedNAR [16]–designed for heterogeneous FL. Building upon each baseline, we incorporate Flick and compare it with **seven** counterparts: methods tailored to mitigate domain shift such as FedBN [26] and FedHEAL [25]; and generative methods including DynaFed [27], FedFTG [19], and FGL [20] (server-side), as well as FRAug [21] and Gen-FedSD [50] (client-side). We run all methods three times per setup and report the average and standard deviation of *Top-1 accuracy*. Model convergence is assessed by the round-to-accuracy performance (*#Round*) defined as the number of rounds required to reach the target accuracy (i.e., the best accuracy of baseline methods). As a generative method, we also report statistics on generated data points, including the number of samples at the target accuracy (*#Sample*) and the total generated samples over all rounds (*#Total Sample*). Evaluation on the temporal scale is provided in Appendix D.

Table 1: Global model accuracy (%) under both domain shift and label skew. AVG denotes the average accuracy across all domains, and $\Delta$ indicates the accuracy gain compared with vanilla methods. **Best** in bold. *Acronyms in the PACS dataset: Photo (P), Art Painting (A), Cartoon (C), and Sketch (S); In the Office-Caltech dataset: Amazon (A), Caltech (C), DSLR (D), and Webcam (W).*

| Methods | PACS | | | | | | | Office-Caltech | | | | | | |
|---|---|---|---|---|---|---|---|---|---|---|---|---|---|---|
| | P | A | C | S | AVG | $\Delta\uparrow$ | #Round$\downarrow$ | A | C | D | W | AVG | $\Delta\uparrow$ | #Round$\downarrow$ |
| *FedAvg* [5] | 73.54 | 86.37 | 84.82 | 87.49 | 83.06$_{\pm1.19}$ | - | 142 | 68.52 | 62.44 | 79.46 | 72.92 | 70.84$_{\pm1.31}$ | - | 135 |
| +FedBN [26] | 80.70 | 84.82 | 90.33 | 90.29 | 86.54$_{\pm0.45}$ | 3.48 | 81 | 66.58 | 62.89 | 82.14 | 80.83 | 73.11$_{\pm0.71}$ | 2.27 | 135 |
| +FedHEAL [25] | 85.19 | 86.91 | 93.45 | 85.96 | 87.88$_{\pm0.14}$ | 4.82 | 50 | 73.23 | 63.56 | 79.76 | 80.56 | 74.28$_{\pm0.96}$ | 3.44 | 48 |
| +DynaFed [27] | 74.39 | 85.03 | 87.95 | 89.34 | 84.18$_{\pm1.46}$ | 1.12 | 79 | 78.91 | 62.45 | 72.10 | 72.34 | 71.45$_{\pm0.72}$ | 0.61 | 99 |
| +FedFTG [19] | 75.61 | 89.81 | 88.10 | 88.64 | 85.54$_{\pm0.17}$ | 2.48 | 90 | 72.69 | 62.89 | 72.10 | 64.00 | 67.92$_{\pm0.80}$ | -2.92 | - |
| +FGL [20] | 88.96 | 87.69 | 97.62 | 85.09 | 89.84$_{\pm0.60}$ | 6.78 | 22 | 67.36 | 62.22 | 85.71 | 80.00 | 73.82$_{\pm1.72}$ | 2.98 | 45 |
| +Flick | 91.99 | 94.59 | 97.74 | 93.91 | **94.49**$_{\pm0.10}$ | **11.43** | **11** | 75.82 | 64.00 | 96.43 | 77.78 | **78.51**$_{\pm0.56}$ | **7.67** | **37** |
| *FedProx* [9] | 74.82 | 87.53 | 86.98 | 87.69 | 84.25$_{\pm1.17}$ | - | 142 | 65.93 | 60.11 | 79.46 | 76.25 | 70.44$_{\pm0.68}$ | - | 135 |
| +FedBN [26] | 77.43 | 87.05 | 88.39 | 89.02 | 85.47$_{\pm0.25}$ | 1.22 | 113 | 64.77 | 62.22 | 82.14 | 78.33 | 71.87$_{\pm0.51}$ | 1.43 | 59 |
| +FedHEAL [25] | 84.10 | 86.84 | 94.79 | 84.20 | 87.48$_{\pm0.17}$ | 3.23 | 50 | 76.68 | 64.89 | 78.57 | 68.33 | 72.12$_{\pm0.54}$ | 1.68 | 59 |
| +DynaFed [27] | 77.43 | 85.77 | 87.05 | 87.94 | 84.55$_{\pm0.49}$ | 0.30 | 81 | 77.65 | 61.11 | 79.24 | 69.00 | 70.25$_{\pm1.33}$ | -0.19 | - |
| +FedFTG [19] | 73.79 | 89.17 | 84.52 | 91.62 | 84.78$_{\pm0.60}$ | 0.53 | 115 | 70.47 | 62.89 | 75.67 | 65.67 | 68.68$_{\pm1.72}$ | -1.76 | - |
| +FGL [20] | 87.99 | 91.51 | 95.68 | 88.13 | 90.83$_{\pm0.24}$ | 6.58 | 21 | 72.54 | 65.56 | 82.14 | 76.67 | 74.23$_{\pm0.11}$ | 3.79 | 69 |
| +Flick | 91.87 | 94.16 | 98.07 | 93.27 | **94.34**$_{\pm0.09}$ | **10.09** | **19** | 74.96 | 64.59 | 96.43 | 79.44 | **78.86**$_{\pm0.47}$ | **8.42** | **40** |
| *FedDyn* [49] | 75.12 | 83.76 | 91.07 | 87.50 | 84.36$_{\pm0.20}$ | - | 22 | 70.98 | 59.11 | 82.14 | 80.83 | 73.27$_{\pm0.54}$ | - | 100 |
| +FedBN [26] | 78.76 | 85.56 | 89.88 | 88.58 | 85.70$_{\pm0.22}$ | 1.34 | 22 | 75.39 | 64.67 | 83.93 | 78.33 | 75.58$_{\pm1.45}$ | 2.21 | 97 |
| +FedHEAL [25] | 80.22 | 85.67 | 91.22 | 86.68 | 85.95$_{\pm0.27}$ | 1.59 | 26 | 74.09 | 60.00 | 78.57 | 86.67 | 74.83$_{\pm0.98}$ | 1.56 | 50 |
| +DynaFed [27] | 78.80 | 84.47 | 91.17 | 83.80 | 85.31$_{\pm0.63}$ | 0.95 | 15 | 75.73 | 60.45 | 77.67 | 74.34 | 72.05$_{\pm0.46}$ | -1.22 | - |
| +FedFTG [19] | 79.05 | 85.56 | 91.77 | 85.66 | 85.51$_{\pm0.55}$ | 1.15 | 20 | 75.47 | 62.07 | 78.14 | 73.78 | 72.37$_{\pm0.52}$ | -0.90 | - |
| +FGL [20] | 81.96 | 87.33 | 91.87 | 84.05 | 86.30$_{\pm0.49}$ | 1.94 | 18 | 72.80 | 60.89 | 82.14 | 80.83 | 74.17$_{\pm0.20}$ | 0.90 | 37 |
| +Flick | 87.50 | 93.42 | 95.24 | 93.40 | **92.39**$_{\pm0.14}$ | **8.03** | **12** | 77.85 | 63.33 | 91.07 | 83.75 | **79.00**$_{\pm0.71}$ | **5.73** | **22** |
| *FedNAR* [16] | 81.19 | 87.47 | 93.60 | 80.52 | 85.70$_{\pm0.46}$ | - | 142 | 67.88 | 60.89 | 76.79 | 78.33 | 70.97$_{\pm0.46}$ | - | 135 |
| +FedBN [26] | 82.77 | 89.17 | 94.49 | 88.96 | 88.85$_{\pm0.82}$ | 3.15 | 35 | 69.95 | 60.22 | 78.57 | 81.67 | 72.60$_{\pm0.48}$ | 1.63 | 129 |
| +FedHEAL [25] | 83.37 | 83.35 | 95.39 | 84.77 | 87.22$_{\pm0.20}$ | 1.52 | 81 | 73.06 | 63.56 | 85.71 | 71.67 | 73.50$_{\pm0.35}$ | 2.90 | 45 |
| +DynaFed [27] | 86.08 | 87.90 | 94.74 | 86.93 | 88.91$_{\pm0.53}$ | 3.21 | 46 | 68.03 | 63.33 | 79.24 | 65.67 | 69.09$_{\pm0.63}$ | -1.88 | - |
| +FedFTG [19] | 80.95 | 91.72 | 93.15 | 86.04 | 87.97$_{\pm0.88}$ | 2.27 | 100 | 78.91 | 65.11 | 61.38 | 65.67 | 67.78$_{\pm1.83}$ | -3.19 | - |
| +FGL [20] | 87.99 | 87.90 | 96.58 | 85.85 | 89.58$_{\pm0.81}$ | 3.88 | 22 | 69.43 | 62.45 | 85.71 | 70.83 | 72.11$_{\pm0.32}$ | 1.14 | 47 |
| +Flick | 92.35 | 91.83 | 97.47 | 92.83 | **93.62**$_{\pm0.31}$ | **7.92** | **12** | 73.96 | 62.22 | 91.96 | 80.42 | **77.14**$_{\pm0.49}$ | **6.17** | **43** |

**Implementation Details.** For fair comparisons, all methods are implemented using the same settings. We use SGD as an optimizer with a learning rate of 0.01; the weight decay is $4e-5$ and the momentum is 0.9. The batch size for local training is 64 and 32 for the two datasets, respectively, with four clients participating in each round. The communication rounds are set to 150. We utilize "Salesforce/blip-image-captioning-large" [24] for the image captioning and "sd-legacy/stable-diffusion-v1-5" [51] for the image generation, sourced from Hugging Face. We also use "gpt-4o-mini" from OpenAI API [52] to analyze offloaded captions. More details are given in Appendix B.

## 4.2 Performance Evaluation

**Flick Effectiveness.** Table 1 presents performance under the heterogeneous setting with both domain shift and label skew. We observe that Flick outperforms all counterpart methods across integrated baselines, consistently yielding substantial improvements in global model accuracy and requiring significantly fewer rounds to reach the target accuracy. Specifically, Flick improves the Top-1 accuracy by up to 11.43% on the PACS dataset and 8.42% on the Office-Caltech dataset while reducing #Round from 142 and 100 to 11 and 22, respectively. The performance of DynaFed and FedFTG heavily depends on the quality of generated pseudo samples, posing significant challenges when facing relatively high-resolution data such as $224\times224\times3$ images adopted in our experiments, resulting in severe variance across the two datasets. FGL [20] constructs a large IID synthetic dataset before FL training using captions from all local data, with its size matched to Flick for a fair comparison, which is used to fine-tune the global model to boost FL performance. However, due to its reliance on local data side information, FGL underperforms compared to Flick. Additionally, creating a large synthetic dataset in advance introduces substantial overhead, making it difficult to reach the target accuracy within a short time frame (see Appendix D). Besides, we investigate the performance of Flick's under the domain shift-only setup where each client holds data from a single domain while maintaining balanced label distributions across clients. Figure 5 shows the superior performance of Flick compared to domain shift-specific methods (i.e., FedBN and FedHEAL) across two datasets. These results highlight that Flick effectively fuses cross-client-specific knowledge while installing commonsense insights from the LLMs into synthetic data, significantly mitigating the heterogeneity problem in FL. More experimental results can be found in Appendix C.2.

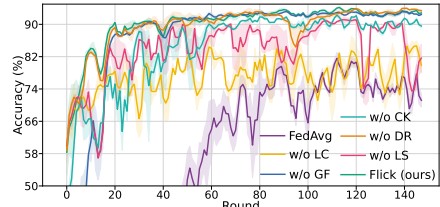

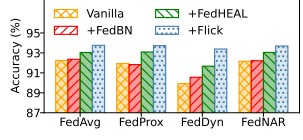

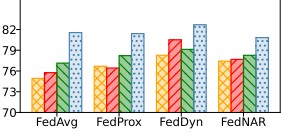

(a) PACS dataset    (b) Office-Caltech dataset

Figure 4: Learning curve of ablation study.

Figure 5: Performance across various baseline methods under domain shift only.

**Comparison with Local Generative Methods.** We also compare Flick with local generative methods: FRAug, Gen-FedSD, and a transformation-based local augmentation method that locally generates data points through random transformations such as cropping, rotation, and image mirroring [50]. For Gen-FedSD and transformation-based methods, the number of samples per class depends on the local data distribution, with more samples generated for underrepresented classes. To ensure a fair comparison, we

Table 2: Performance across local generative methods.

| Methods | PACS | | | | | |
|---|---|---|---|---|---|---|
| | P | A | C | S | AVG ↑ | #Round ↓ |
| FRAug [21] | 79.85 | 85.99 | 90.48 | 84.39 | $85.18_{\pm0.32}$ | 21 |
| Transformation [50] | 87.94 | 90.16 | 95.04 | 90.57 | $90.93_{\pm0.52}$ | 31 |
| Gen-FedSD [22] | 90.78 | 94.27 | 96.73 | 93.15 | $93.73_{\pm0.27}$ | 19 |
| **Flick** | 91.99 | 94.59 | 97.74 | 93.91 | $\mathbf{94.49_{\pm0.10}}$ | **11** |

| Methods | Office-Caltech | | | | | |
|---|---|---|---|---|---|---|
| | A | C | D | W | AVG ↑ | #Round ↓ |
| FRAug [21] | 58.03 | 50.67 | 82.14 | 85.00 | $68.96_{\pm0.66}$ | - |
| Transformation [50] | 72.88 | 62.52 | 75.00 | 77.22 | $71.91_{\pm1.84}$ | 138 |
| Gen-FedSD [22] | 77.98 | 64.44 | 83.93 | 75.00 | $75.34_{\pm0.21}$ | 48 |
| **Flick** | 75.82 | 64.00 | 96.43 | 77.78 | $\mathbf{78.51_{\pm0.56}}$ | **37** |

keep the consistent budget of generated samples across all methods with Flick. Table 2 shows that local generative methods, by leveraging their own determined data distributions, can perform precise augmentation to balance local datasets effectively. This leads to notable performance gains over other model-driven approaches and even some server-side methods. Among them, sample-wise methods (i.e., Gen-FedSD and Transformation) consistently outperform the feature-wise method (i.e., FRAug) across two benchmarks. Although Flick is a server-side data-driven framework, it avoids introducing data generation overhead on clients while achieving superior performance by centralizing LLMs usage and leveraging cross-client-specific knowledge.

**Ablation Study.** To assess the effectiveness of key components in Flick, we conduct an ablation study on the PACS dataset with five breakdowns: *Local Summary (LS), Commonsense Knowledge (CK), Data Retrieval (DR), Local Compensation (LC), and Global Fine-tuning (GF)*. Results in Table 3 show that the variant of Flick without the local summary phase leads to a 2.67% accuracy drop, as the server generates the data solely based on class names. It highlights the importance of

Table 3: Ablation study on the PACS dataset.

| LS | CK | DR | LC | GF | AVG | #Round | #Sample | #TotalSample |
|---|---|---|---|---|---|---|---|---|
| ✗ | ✔ | ✔ | ✔ | ✔ | $91.82_{\pm0.29}$ | 19 | 430 | 1218 |
| ✔ | ✗ | ✔ | ✔ | ✔ | $92.92_{\pm0.22}$ | 27 | 312 | **400** |
| ✔ | ✔ | ✗ | ✔ | ✔ | $\mathbf{94.75_{\pm0.13}}$ | **11** | 339 | 1665 |
| ✔ | ✔ | ✔ | ✗ | ✔ | $89.11_{\pm0.40}$ | 26 | 630 | 3239 |
| ✔ | ✔ | ✔ | ✔ | ✗ | $93.75_{\pm0.04}$ | 19 | 504 | 1169 |
| ✔ | ✔ | ✔ | ✔ | ✔ | $94.49_{\pm0.10}$ | 11 | **270** | 1083 |

cross-client-specific knowledge in acquiring task-related information from LLMs. Flick without commonsense knowledge directly uses client-specific captions for data generation, relying solely on the local summary. As training progresses, local compensation depends primarily on retrieved samples rather than high-quality data generated through the integration of cross-client insights and commonsense knowledge, leading to fewer generated samples, slower convergence, and reduced model accuracy. The variant of Flick without local compensation brings more data generation overhead while introducing limited performance improvement. Besides, Flick without data retrieval does not significantly affect performance; however, it results in more samples being generated, introducing extra overhead without proportional gains. Figure 4 further showcases the benefits of synthetic data usage: global fine-tuning speeds up the model convergence at the beginning, and the combination of local compensation and global fine-tuning yields consistent and substantial performance improvements throughout training. More ablation studies can be found in Appendix C.3.

**Scalability Analysis.** To further evaluate the scalability of Flick in large-scale federated learning, we extend our analysis to the DomainNet dataset [45] with 100 clients. As illustrated in Figure 6, Flick consistently delivers superior performance in both model accuracy and round-to-accuracy (#Round) across four different baseline methods, demonstrating its robustness under highly heterogeneous and

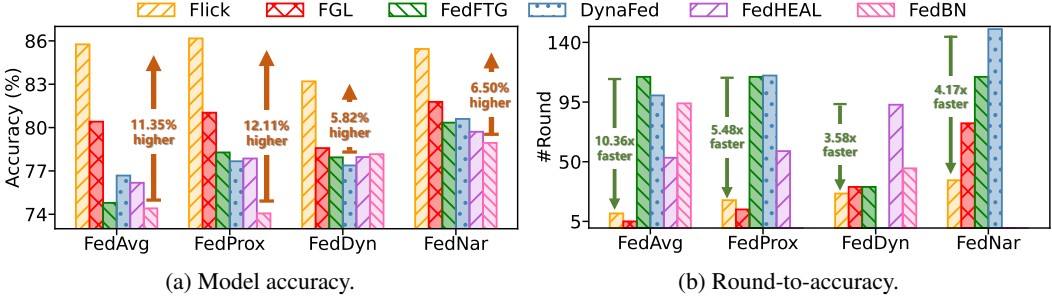

| (a) Model accuracy. | (b) Round-to-accuracy. |

Figure 6: Model performance across various baseline methods on the DomainNet dataset.

large-population FL scenarios. In particular, Flick improves accuracy by up to 11.35%, 12.11%, 5.82%, and 6.50%, and accelerates convergence by up to 10.36×, 5.48×, 3.58×, and 4.17× compared with the respective baselines. Notably, although FGL achieves fast convergence by fine-tuning the global model on a large IID synthetic dataset, this comes at the expense of substantial computational and generative overhead, making Flick more appealing in resource-constrained federated settings. Comprehensive configurations and detailed results are included in Appendix C.4.

## 5 Discussion

We provide extensive quantitative analyses in the Appendix D, E, F, offering deeper insights into Flick's system efficiency, privacy preservation, and practical deployability. Flick improves global performance without exposing raw data, as only lightweight and low-sensitivity token summaries are transmitted, inherently reducing privacy risks. Although server-side synthetic data generation introduces additional computation, this cost is fully offloaded from clients and is largely offset by the fast convergence enabled through rich LLM knowledge infusion. Moreover, directly deploying large models on clients is infeasible in real-world edge intelligence due to strict resource constraints, and querying cloud-based LLMs could cause privacy leakage. In contrast, Flick allows compact client models to inherit the knowledge of LLMs in a cost-effective and communication-efficient manner. We believe Flick highlights that foundation models and lightweight client models play complementary roles: while large models provide rich knowledge, small models remain the only practical and privacy-preserving learners at the edge, making their collaboration essential in real deployments.

## 6 Conclusion

In this paper, we design a novel generative framework Flick to address the data heterogeneity, including both label skew and domain shift issues in FL, where the central server generates synthetic samples by integrating cross-client-specific knowledge with the commonsense knowledge embedded in out-of-the-box LLMs. Flick allows clients to selectively caption their samples, distilling client-specific knowledge into low-sensitivity textual information that is then explored by the server to generate data with the power of LLMs to boost FL training. We also explore the generated data for local data compensation and global model fine-tuning, further enhancing the global model accuracy and convergence performance. Extensive experiments on three datasets have been conducted to validate the effectiveness of Flick. Specifically, Flick improves global model accuracy by up to 11.43% and facilitates round-to-accuracy performance from 142 to 11. Besides, Flick produces consistently superior performance against other counterparts on DomainNet with 100 clients, demonstrating its potential to enhance model performance in large-scale heterogeneous scenarios.

## Acknowledgement

This work is partly funded by the EU's Horizon Europe HarmonicAI project under the HORIZON-MSCA-2022-SE-01 scheme with grant agreement number 101131117. The work is also part of the ICAI GENIUS lab of the research program ROBUST (project number KICH3.LTP.20.006), partly funded by the Dutch Research Council (NWO). We also thank the support provided by Samenwerkende Universitaire RekenFaciliteiten (SURF) with their Snellius infrastructure.

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

# Appendix

## A  Notations

All notations used in this paper are summarized in Table 4. We omit the superscript of symbols indicating temporal information, e.g., $\mathcal{J}^n$ refers to a set of participating clients in communication round $n$.

Table 4: Symbols and notations in the paper.

| Symbol | Explanation |
| --- | --- |
| $\mathcal{J}$ | Set of clients |
| $\mathcal{D}_j, \hat{\mathcal{D}}_j$ | Set of local/representative samples of client $j$ |
| $\tilde{\mathcal{D}}_j$ | Set of generated samples for client $j$ |
| $\tilde{\mathcal{T}}_j, \mathcal{T}_j$ | Tokens sequences/text captions from client $j$ |
| $\mathcal{G}_s$ | Data pool maintained by server |
| $\mathcal{M}_i$ | Set of text prompts for class $i$ |
| $\mathcal{W}_j$ | Weights of updated local model |
| $\bar{\mathcal{W}}, \mathcal{W}$ | Weights of aggregated/fine-tuned global model |
| $\mathbf{D}$ | Decision matrix for data generation |
| $\mathcal{I}$ | Set of classes of generated samples |
| $\tau_j$ | Number of local SGD steps for client $j$ |
| $N$ | Number of communication rounds |
| $C$ | Number of task-specific classes |
| $G$ | Budget for the number of generated samples |
| $T_v, T_s$ | Threshold for validation accuracy/text similarity |

## B  Experimental Details

### B.1  Visualization of the PACS, Office-Caltech, and DomainNet datasets

We evaluate the performance of Flick and its counterparts on three benchmark image datasets: PACS [41], Office-Caltech [42], and DomainNet [45]. Each benchmark is a multi-class classification task, where each class includes samples from different domains. The PACS dataset comprises seven classes: *dog, elephant, giraffe, guitar, horse, house, and person*. The Office-Caltech dataset contains

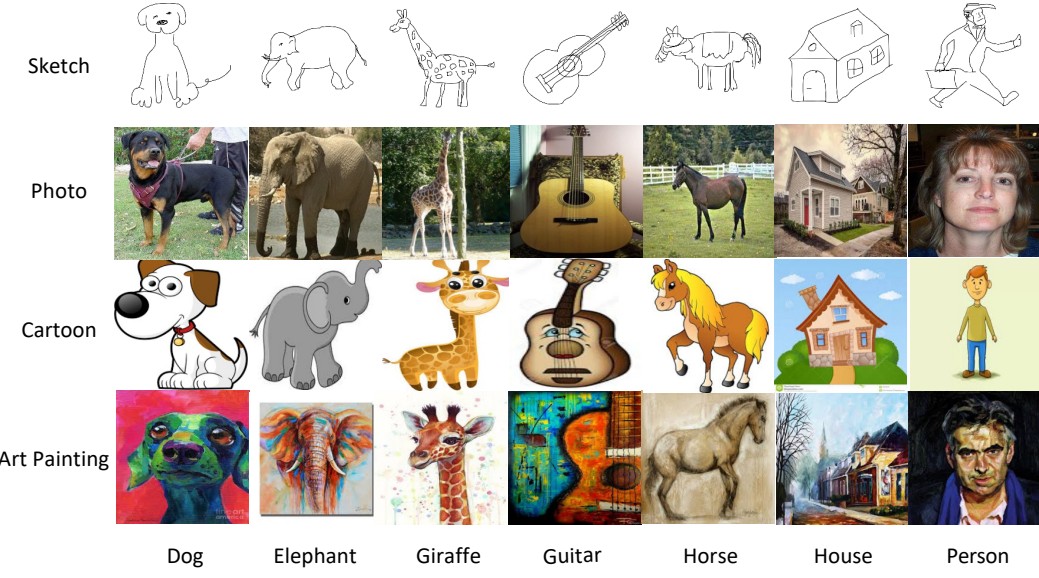

Figure 7: Example samples from different classes and domains in the PACS dataset [41].

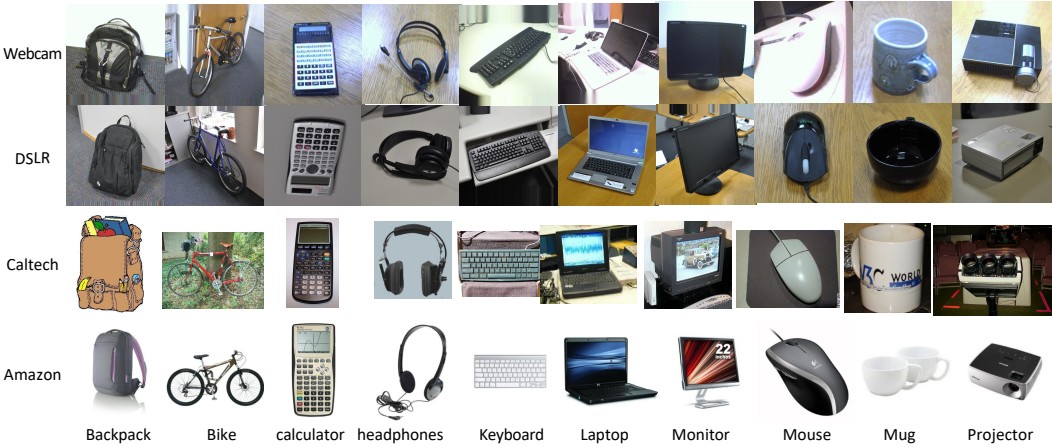

Figure 8: Example samples from different classes and domains in the Office-Caltech dataset [42].

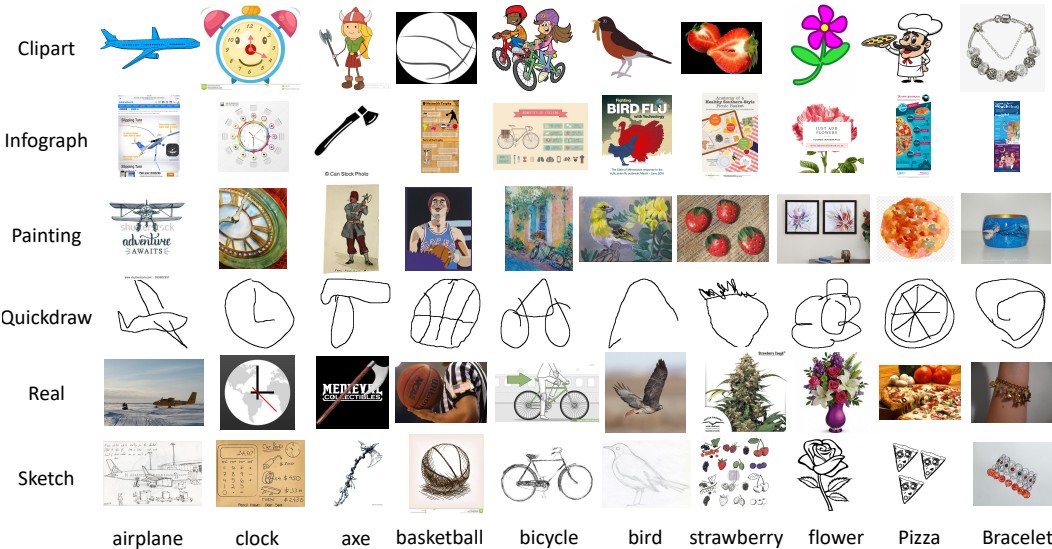

Figure 9: Example samples from different classes and domains in the DomainNet dataset [45].

ten classes: *backpack, bike, calculator, headphones, keyboard, laptop, monitor, mouse, mug, and projector*. Following prior work [20, 26, 46], we construct a sub-dataset of the DomainNet dataset for our experiments by selecting the top ten most common classes: *airplane, clock, axe, basketball, bicycle, bird, strawberry, flower, pizza, bracelet*. Figures 7, 8, and 9 illustrate the diverse domains within individual classes. In the PACS dataset, significant feature variations across domains within the same class are clearly observable, making domain differences explicit. In contrast, domain differences in Office-Caltech are more latent, for instance, variations with background, view, and clarity of the object, posing additional challenges in effectively addressing the domain shift problem within this benchmark. DomainNet contains six highly heterogeneous domains, where the appearance gap of the same class across domains can be extremely large, making our experiments on this dataset more realistic and representative of real-world scenarios. To simulate heterogeneous FL settings with domain shifts, we partition the data and assign data from one domain exclusively to each client.

## B.2 Hyperparameter Settings

Here, we provide more details about the experimental training parameter settings. All methods are implemented in Python, with neural networks developed using PyTorch. In local training, local epochs are set to 5, and the input data size is fixed at $224 \times 224$. We set the server-held data pool size

$|\mathcal{G}_s|$ to 25, and the data generation budget $G$ to 5 for both datasets. We use "stable-diffusion-v1-5" for the image generator and "gpt-4o-mini" and out-of-the-box LLMs for generating text prompts by analyzing the uploaded client-specific knowledge combined with embedded commonsense knowledge. A comprehensive robustness evaluation of Flick under different image generators and out-of-the-box LLMs is provided in Appendix C.1. Additionally, for the PACS dataset, we set text similarity threshold $T_s = 0.8$ and validation threshold $T_v = 0.9$, which guide the construction of the decision matrix $\mathbf{D}$. For the Office-Caltch dataset, we adopt $T_s = 0.7$ and $T_v = 0.8$ as the default settings.

In baseline methods, FedProx uses a hyperparameter to control the proximal term's weight in the objective function. We tune this hyperparameter within the $[0.001, 0.01, 0.1, 1]$ following [9], and report the best result. Specifically, the value of 0.1 is adopted in both datasets. For FedDyn, we provide its best performance by fine-tuning the weight of penalized risk within $[0.001, 0.01, 0.1]$. We set this value to 0.1 for PACS and 0.01 for Office-Caltech. For FGL, we adopt the multi-round-syn variant, where the global model is initially trained on a synthetic dataset generated in the first round and subsequently fine-tuned using synthetic data throughout the following communication rounds. To ensure a fair comparison, we match the total amount of synthetic data to that produced by our method, Flick, under each experimental setting. For other counterparts, we keep the default settings they provided.

## C  More Results and Analysis

### C.1  Sensitivity Analysis

In this section, we demonstrate that Flick consistently maintains superior performance across a range of hyperparameter settings and model choices, indicating the robust generalization of Flick without reliance on extensive tuning.

**Impact of text similarity $T_s$, validation accuracy $T_v$, and data pool size.** Figure 10(a) shows the model performance improvements with a larger text similarity threshold $T_s$. The number of generated samples required to reach the target accuracy (i.e., FedAvg's best accuracy) rises with a higher $T_s$, while #Sample decreases because fewer rounds are needed. When $T_s$ reaches 0.9, no samples are retrieved. For validation accuracy threshold $T_v$, a lower value generates fewer samples per round but requires more rounds to reach the target accuracy. With a fixed number of rounds, the accuracy is lower at smaller values of $T_v$. As shown in Figure 10(b), $T_v = 0.9$ or 1.0 serves as an effective threshold, yielding better performance with fewer required samples. Figure 10(c) shows that the accuracy performance does not change much across different sizes of the server-held data pool. Here, x-axis denotes storage capacity per class in the data pool; we can observe that a larger pool size significantly benefits the global model's fine-tuning, greatly reducing #Sample. However, a larger data pool size requires more storage.

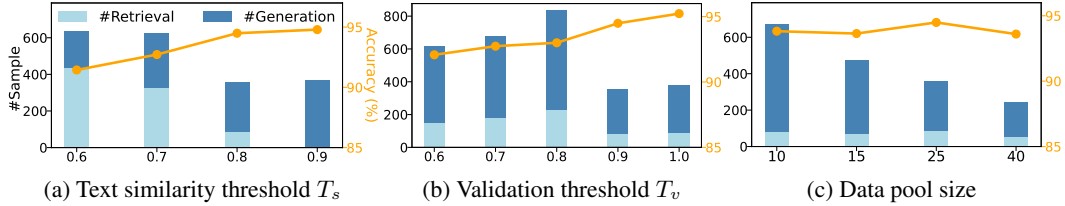

(a) Text similarity threshold $T_s$     (b) Validation threshold $T_v$     (c) Data pool size

Figure 10: The performance of Flick under varying hyperparameters on the PACS dataset.

**Impact of Data Generation Budget $G$.** In our designed prompt illustrated in Figure 3, the server hosting the LLM can generate text prompts of varying sizes for class $i$ under a specified hyperparameter $G$. This design flexibility in our Flick allows the server to adapt to the practical deployment with the given data generation budget. In the experiments, we evaluate the performance of Flick under different data generation budgets $G \in \{5, 10, 15\}$. We keep the experimental setup for the two benchmarks consistent with the experiments in Table 1, with FedAvg as the baseline method, except for varying $G$ and the server-held data pool size fixed at $|\mathcal{G}_s| = 40$. The results are shown in Table 5. We can observe that a larger $G$ indeed improves the global model accuracy in the long run by generating more samples enriched with commonsense knowledge. Additionally, $G = 10$ and $G = 15$ speed up the model convergence on the Office-Caltech dataset compared to $G = 5$, as evidenced by

Table 5: The performance of Flick under different data generation budget $G$ on both datasets.

| $G$ | PACS | | | | | | |
| --- | --- | --- | --- | --- | --- | --- | --- |
| | P | A | C | S | AVG ↑ | #Round ↓ | #Sample ↓ |
| $G = 5$ | 91.99 | 91.51 | 97.62 | 93.27 | $93.60_{\pm 0.17}$ | 7 | 188 |
| $G = 10$ | 92.72 | 94.06 | 96.43 | 93.65 | $94.21_{\pm 0.20}$ | 7 | 276 |
| $G = 15$ | 93.93 | 93.84 | 97.17 | 93.53 | $94.62_{\pm 0.14}$ | 7 | 420 |
| $G$ | Office-Caltech | | | | | | |
| | A | C | D | W | AVG ↑ | #Round ↓ | #Sample ↓ |
| $G = 5$ | 76.68 | 65.78 | 88.10 | 80.56 | $77.78_{\pm 0.12}$ | 33 | 146 |
| $G = 10$ | 77.98 | 63.56 | 94.64 | 79.17 | $78.84_{\pm 0.22}$ | 24 | 253 |
| $G = 15$ | 77.72 | 65.33 | 96.43 | 78.33 | $79.45_{\pm 0.18}$ | 24 | 317 |

the enhanced round-to-accuracy performance (i.e., #Round). However, $G = 5$ requires the fewest generated samples (i.e., #Sample) to reach the target accuracy, revealing a trade-off between training performance (e.g., accuracy and latency) and data generation overheads. It underscores the importance of carefully selecting the $G$ to balance data generation costs and performance improvements in practical applications. In the rest of the evaluations in this paper, we use $G = 5$ as the default setting.

**Impact of different image captioning methods.** In Flick, clients perform image captioning on a small set of representative local samples during the summary phase. This summarization captures essential local insights that play a crucial role in guiding the server to extract task-relevant, high-quality commonsense knowledge from LLMs – an effect validated in our ablation study. To assess the sensitivity of Flick to different image captioning strategies, we evaluate its performance using three representative captioning models: "blip-image-captioning-large", "blip-image-captioning-base", and "vit-gpt2-image-captioning". As shown in Figure 11, the results on both datasets indicate that Flick is agnostic to the choice of captioning model and consistently maintains superior performance. This demonstrates the flexibility and generalization of our approach.

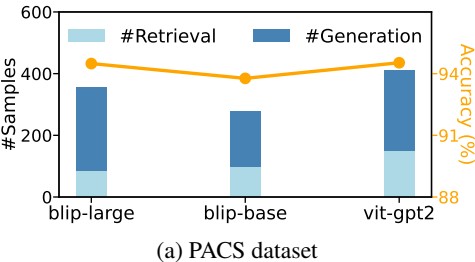
(a) PACS dataset

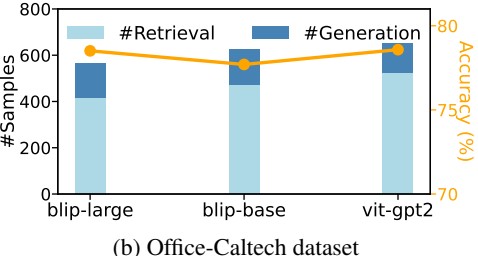
(b) Office-Caltech dataset

Figure 11: The performance of Flick over different image captioning methods: blip-image-captioning-large, blip-image-captioning-base, and vit-gpt2-image-captioning.

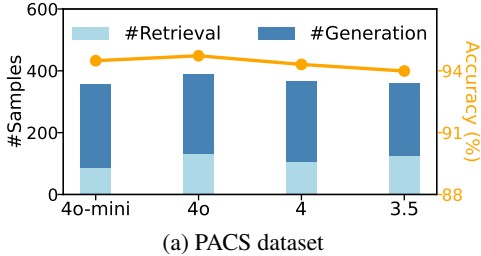
(a) PACS dataset

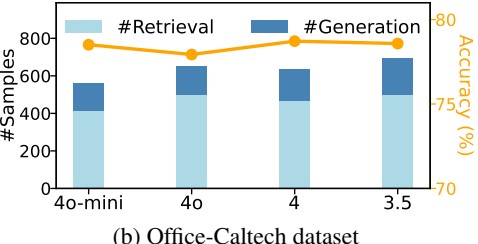
(b) Office-Caltech dataset

Figure 12: The performance of Flick over different out-of-the-box LLMs: gpt-4o-mini, gpt-4o, gpt-4, and gpt-3.5-turbo-0125.

**Impact of different out-of-the-box LLMs.** The server in Flick employs LLM to analyze the collected local summary and generate text prompts for subsequent data generation. The designed prompt template instructs the LLM to extract class-specific information from collected local captions

and fuse it with inherent commonsense knowledge, thereby enhancing the quality of the output text prompts. To demonstrate the compatibility of the Flick workflow with out-of-the-box LLMs, we use the same prompt template as illustrated in Figure 3 and evaluate the performance of Flick by configuring the LLM to "gpt-3.5-turbo-0125", "gpt-4", "gpt-4o", and "gpt-4o-mini" (default LLM in this paper). The experimental results are given in Figure 12. We can see that the choice of LLM has a negligible impact on the text prompt generation, evidenced by the similar global model performance and data generation decisions. These valuation results validate the compatibility of our designed prompt template across a wide range of LLMs.

**Impact of Image Generator** $f^{\mathcal{W}_G}(\cdot)$. As a key component of the Flick workflow, a text-to-image model is employed to generate synthetic samples taking input text prompts. Since the data generation runs on the server side, the availability of rich computational resources allows for extending the choice within advanced generative image models, thereby enabling the generation of high-quality data points for further usage in Flick. To this end, Figure 13 reports the performance of Flick when using various generative models. We investigate the LDMs publicly available in HuggingFace, including "stable-diffusion-v1-5" (default model in this paper), "stable-diffusion-v1-4", and "stable-diffusion-xl-base-1.0". We also incorporate the "DALL-E-2" embedded in ChatGPT/OpenAI API for comparison. The results indicate that synthetic samples provided by all four image generators significantly enhance FL performance compared to the baseline FedAvg that achieves 83.06% on PACS and 70.84% on Office-Caltech. Note that the models with more advanced designs improve the global model's performance better, at the cost of higher computational demands. E.g., "DALL-E-2" model leads to higher accuracy, potentially due to its ability to produce samples in diverse artistic styles [53].

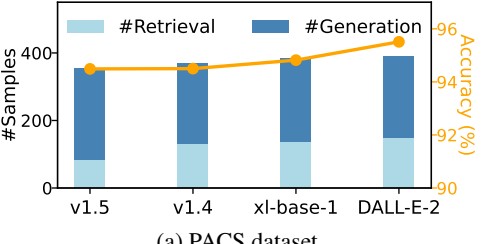
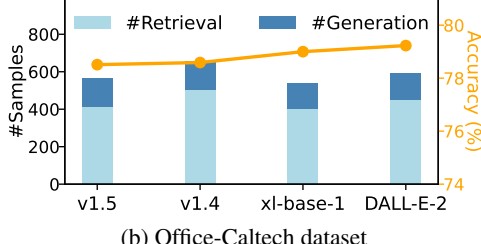

(a) PACS dataset

(b) Office-Caltech dataset

Figure 13: The performance of Flick over different image generators: stable-diffusion-v1-5, stable-diffusion-v1-4, stable-diffusion-xl-base-1.0, and DALL-E-2.

## C.2 Detailed Evaluation under Domain Shift

The data generation process in Flick considers the class-wise performance of local models (i.e., through the decision matrix $\mathbf{D}$) and the domain knowledge variations across clients (i.e., the collection of local summaries $\mathcal{T}$ used in LLM prompt). This highlights the capability of Flick to solve domain shift and label skew in heterogeneous FL jointly and therefore distinguishes Flick from other model-driven [49, 16, 9] or data-driven [17, 18, 21, 22] methods that solely address imbalanced local datasets across clients. To validate the performance gain by Flick attributed to mitigating domain shift, Table 6 reports experimental results where each client holds balanced local data in label space but only from one domain. It shows that Flick consistently yields significant improvement in model accuracy and reduces the number of communication rounds (#Round) to reach the target accuracy–defined as the best accuracy of the baseline methods (i.e., FedAvg, FedProx, FedDyn, FedNAR). Among generation-based counterparts, DynaFed and FedFTG have limited performance gain under the heterogeneous settings with domain shift alone and even show inferior performance to the baselines on the Office-Caltech dataset. For FGL, we match the synthetic data volume to that of Flick. For example, under the FedAvg baseline on the PACS dataset, Flick generates only 300 synthetic samples in total. This limited amount of auxiliary data constrains the performance of FGL – a finding consistent with its released paper. In contrast, the same amount proves sufficient for Flick to deliver significant performance improvements. More importantly, compared to methods specifically designed to address domain shift (i.e., FedBN and FedHEAL), Flick further improves the global model performance and convergence by a large margin. Combined with the results from Table 1, it is clear that Flick shows great performance in addressing data heterogeneity issues regarding label skew and domain shift.

Table 6: Global model accuracy performance (%) under domain shift only. AVG denotes the average accuracy calculated across all domains, and $\Delta$ indicates the accuracy gain compared with vanilla methods. **Best** in bold. *Acronyms in the PACS dataset: Photo (P), Art Painting (A), Cartoon (C), and Sketch (S); In the Office-Caltech dataset: Amazon (A), Caltech (C), DSLR (D), and Webcam (W).*

| Methods | PACS | | | | | | | Office-Caltech | | | | | | |
|---|---|---|---|---|---|---|---|---|---|---|---|---|---|---|
| | P | A | C | S | AVG | $\Delta\uparrow$ | #Round$\downarrow$ | A | C | D | W | AVG | $\Delta\uparrow$ | #Round$\downarrow$ |
| *FedAvg* [5] | 91.85 | 94.16 | 95.98 | 87.88 | $92.47_{\pm0.11}$ | - | 84 | 77.47 | 63.33 | 70.37 | 88.56 | $74.93_{\pm0.70}$ | - | 59 |
| +FedBN [26] | 89.53 | 93.35 | 95.63 | 90.91 | $92.36_{\pm0.1}$ | -0.13 | - | 77.60 | 64.22 | 72.22 | 88.98 | $75.76_{\pm0.35}$ | 0.83 | 45 |
| +FedHEAL [25] | 91.00 | 94.69 | 95.09 | 91.37 | $93.04_{\pm0.22}$ | 0.57 | 79 | 76.43 | 65.11 | 75.93 | 91.10 | $77.14_{\pm1.18}$ | 2.21 | 38 |
| +DynaFed [27] | 90.27 | 92.71 | 95.93 | 89.68 | $92.15_{\pm0.45}$ | -0.32 | - | 79.69 | 65.33 | 70.37 | 84.75 | $75.03_{\pm0.94}$ | 0.10 | 55 |
| +FedFTG [19] | 89.78 | 92.99 | 90.77 | 92.39 | $91.48_{\pm0.63}$ | -0.99 | - | 82.29 | 65.00 | 67.96 | 78.03 | $73.32_{\pm0.90}$ | -1.61 | - |
| +FGL [20] | 89.90 | 90.13 | 95.09 | 89.97 | $91.27_{\pm0.05}$ | -1.20 | - | 61.04 | 62.22 | 85.19 | 81.19 | $72.41_{\pm0.40}$ | -2.52 | - |
| +Flick | 91.48 | 94.48 | 96.88 | 92.20 | $\mathbf{93.76}_{\pm0.07}$ | **1.29** | **24** | 81.60 | 65.48 | 87.65 | 91.53 | $\mathbf{81.56}_{\pm0.54}$ | **6.63** | **21** |
| *FedProx* [9] | 88.56 | 93.63 | 94.05 | 91.50 | $91.94_{\pm0.25}$ | - | 84 | 78.52 | 62.56 | 75.00 | 90.68 | $76.69_{\pm0.34}$ | - | 59 |
| +FedBN [26] | 89.05 | 93.74 | 94.94 | 89.59 | $91.83_{\pm0.14}$ | -0.11 | - | 74.35 | 64.11 | 78.70 | 88.56 | $76.43_{\pm1.65}$ | -0.26 | - |
| +FedHEAL [25] | 90.88 | 95.12 | 95.39 | 90.99 | $93.09_{\pm0.11}$ | 1.15 | 41 | 79.95 | 67.33 | 74.07 | 91.53 | $78.22_{\pm0.71}$ | 1.53 | 66 |
| +DynaFed [27] | 90.02 | 91.44 | 95.04 | 89.97 | $91.62_{\pm0.45}$ | -0.32 | - | 82.30 | 64.44 | 72.22 | 80.51 | $74.87_{\pm1.67}$ | -1.82 | - |
| +FedFTG [19] | 87.23 | 93.95 | 91.52 | 89.21 | $90.48_{\pm0.34}$ | -1.46 | - | 81.25 | 64.89 | 74.26 | 81.27 | $75.42_{\pm1.23}$ | -1.27 | - |
| +FGL [20] | 90.43 | 89.53 | 94.94 | 89.47 | $91.09_{\pm0.31}$ | -0.85 | - | 68.77 | 63.78 | 85.19 | 84.27 | $76.00_{\pm0.15}$ | -0.69 | - |
| +Flick | 91.36 | 94.27 | 95.83 | 93.46 | $\mathbf{93.73}_{\pm0.16}$ | **1.79** | **16** | 80.21 | 65.33 | 85.19 | 94.92 | $\mathbf{81.41}_{\pm0.15}$ | **4.72** | **16** |
| *FedDyn* [49] | 83.94 | 91.93 | 92.11 | 91.75 | $89.93_{\pm0.44}$ | - | 22 | 79.17 | 62.44 | 83.33 | 88.14 | $78.27_{\pm0.12}$ | - | 68 |
| +FedBN [26] | 86.78 | 91.30 | 92.66 | 91.50 | $90.56_{\pm0.50}$ | 0.63 | 22 | 81.64 | 65.44 | 84.26 | 90.68 | $80.51_{\pm0.43}$ | 2.24 | 79 |
| +FedHEAL [25] | 87.59 | 92.57 | 94.94 | 91.56 | $91.67_{\pm0.10}$ | 1.74 | **16** | 73.44 | 61.11 | 87.04 | 94.92 | $79.13_{\pm0.26}$ | 0.86 | 73 |
| +DynaFed [27] | 87.75 | 91.86 | 94.25 | 90.61 | $91.12_{\pm0.29}$ | 1.19 | 20 | 79.67 | 64.67 | 76.89 | 91.11 | $78.09_{\pm0.71}$ | -0.18 | - |
| +FedFTG [19] | 86.62 | 93.10 | 93.45 | 91.62 | $91.20_{\pm0.47}$ | 1.27 | 20 | 80.73 | 65.44 | 80.18 | 84.14 | $77.62_{\pm0.69}$ | -0.65 | - |
| +FGL [20] | 88.69 | 93.63 | 92.86 | 92.01 | $91.79_{\pm0.06}$ | 1.86 | **13** | 82.29 | 67.85 | 82.72 | 93.22 | $81.52_{\pm0.76}$ | 3.25 | 89 |
| +Flick | 90.02 | 95.01 | 95.39 | 93.15 | $\mathbf{93.39}_{\pm0.33}$ | **3.46** | 16 | 84.11 | 65.56 | 87.03 | 94.07 | $\mathbf{82.69}_{\pm0.59}$ | **4.42** | 23 |
| *FedNAR* [16] | 89.86 | 92.92 | 96.03 | 89.81 | $92.16_{\pm0.13}$ | - | 84 | 77.99 | 63.00 | 75.93 | 92.80 | $77.43_{\pm0.86}$ | - | 66 |
| +FedBN [26] | 90.75 | 92.57 | 94.94 | 90.61 | $92.22_{\pm0.34}$ | 0.06 | 142 | 80.90 | 64.00 | 76.54 | 89.27 | $77.68_{\pm0.70}$ | 0.25 | 131 |
| +FedHEAL [25] | 91.24 | 95.12 | 95.24 | 90.61 | $93.05_{\pm0.47}$ | 0.89 | 30 | 80.21 | 63.56 | 77.78 | 91.53 | $78.27_{\pm0.64}$ | 0.84 | 59 |
| +DynaFed [27] | 89.90 | 92.94 | 94.72 | 89.88 | $91.86_{\pm0.39}$ | -0.30 | - | 81.52 | 67.56 | 74.04 | 88.81 | $77.98_{\pm0.72}$ | 0.55 | 101 |
| +FedFTG [19] | 88.81 | 93.84 | 91.96 | 89.72 | $91.08_{\pm0.72}$ | -1.08 | - | 82.73 | 65.11 | 73.81 | 81.34 | $75.75_{\pm1.06}$ | -1.68 | - |
| +FGL [20] | 90.39 | 90.76 | 95.09 | 89.78 | $91.51_{\pm0.12}$ | -0.65 | - | 72.99 | 64.33 | 83.33 | 84.34 | $76.25_{\pm0.10}$ | -1.18 | - |
| +Flick | 92.34 | 93.74 | 95.98 | 92.70 | $\mathbf{93.69}_{\pm0.18}$ | **1.53** | **16** | 80.38 | 65.33 | 88.89 | 88.70 | $\mathbf{80.83}_{\pm0.54}$ | **3.40** | 25 |

Table 7: Extended ablation study on the components of client-side local summary and server-side data generation in Flick. We compare Flick with its variants: Flick with random local sample captioning (*Random Captioning*) and Flick with random historical sample retrieval (*Random Retrieval*).

| Methods | PACS | | | | | | |
|---|---|---|---|---|---|---|---|
| | P | A | C | S | AVG$\uparrow$ | #Round$\downarrow$ | #Sample$\downarrow$ |
| Random Retrieval | 89.81 | 91.40 | 96.28 | 93.21 | $92.67_{\pm0.07}$ | 19 | 423 |
| Random Captioning | 91.87 | 91.61 | 97.32 | 93.08 | $93.47_{\pm0.33}$ | 19 | 430 |
| **Flick** | 91.99 | 94.59 | 97.74 | 93.91 | $\mathbf{94.49}_{\pm0.10}$ | **11** | **270** |
| Methods | Office-Caltech | | | | | | |
| | A | C | D | W | AVG$\uparrow$ | #Round$\downarrow$ | #Sample$\downarrow$ |
| Random Retrieval | 78.24 | 65.78 | 89.29 | 73.33 | $76.66_{\pm0.52}$ | 45 | 295 |
| Random Captioning | 75.13 | 61.33 | 92.86 | 79.17 | $77.12_{\pm0.20}$ | 43 | 235 |
| **Flick** | 75.82 | 64.00 | 96.43 | 77.78 | $\mathbf{78.51}_{\pm0.56}$ | **37** | **150** |

## C.3 Extended Evaluation on Flick Variants

In Table 3, we conduct the ablation study on Flick by evaluating its performance without the *Local Summary (LS)* phase or the *Data Retrieval (DR)* during the data generation phase. To further explore the effectiveness of our LS and DR designs – specifically the loss-based LS and similarity-based DR strategies – we compare Flick with its variants: random local sample captioning (*Random Captioning*) and random historical sample retrieval (*Random Retrieval*). Table 7 summarizes the results under the setup of FedAvg as the baseline method for both benchmarks. For a fair comparison, the server in the *Random Retrieval* retrieves the same number of samples as Flick but randomly selects from the server-held data pool. This ensures that *Random Retrieval* maintains the same proportion of retrieved samples $\mathcal{D}_{retr}$ and generated samples $\mathcal{D}_{gen}$ as Flick during the entire data generation phase across all communication rounds. The experimental results clearly reveal the benefits of selective local sample summarization and guided historical sample retrieval in Flick over their random counterparts.

Random local sample captioning results in inefficient extraction of client-specific knowledge, as it lacks the focus on captioning the representative local samples provided by the loss-based criteria. Furthermore, the negative impact of introducing randomness into data retrieval is even more severe in *Random Retrieval*, where the central server randomly sends diverse synthetic samples that fail to align with the specific requirements of local data compensation, thereby degrading the training efficiency.

## C.4 Scalability Analysis on DomainNet Dataset

To evaluate the scalability of Flick in large-scale federated settings, we experiment on the DomainNet dataset [45] with 100 clients, where 20% are randomly selected to participate in each communication round. The DomainNet dataset consists of natural images with six different domains: Clipart, Infograph, Painting, Quickdraw, Real, and Sketch. Following prior work [20, 26, 46], we construct a sub-dataset for our experiments by selecting the top ten most common classes: airplane, clock, axe, basketball, bicycle, bird, strawberry, flower, pizza, bracelet. We partition each domain into 15 or 17 subsets using a Dirichlet distribution with concentration parameter $\alpha = 0.1$. Each client receives data from a single domain with a skewed label distribution, simulating a scenario with both label skew and domain shift. We fix the synthetic data volume for FGL at 1500, while the total number of generated samples (#Total Sample) in Flick is lower – 1457, 1133, 1412, and 1299 for FedAvg, FedProx, FedDyn, and FedNova, respectively. Besides, the number of samples required to reach the target accuracy is only 427, 414, 417, and 477, highlighting that Flick achieves superior performance with substantially less generative overhead.

Table 8: Global model accuracy performance (%) under both domain shift and label skew on DomainNet dataset. AVG denotes the average accuracy calculated across all domains, and $\Delta$ indicates the accuracy gain compared with vanilla methods. **Best** in bold.

| Methods | DomainNet | | | | | | | | |
| --- | --- | --- | --- | --- | --- | --- | --- | --- | --- |
| | Clipart | Infograph | Painting | Quickdraw | Real | Sketch | AVG | $\Delta \uparrow$ | #Round $\downarrow$ |
| *FedAvg* [5] | 82.59 | 46.99 | 71.60 | 77.63 | 85.95 | 80.00 | 74.13$_{\pm 0.38}$ | - | 144 |
| +FedBN [26] | 83.64 | 52.00 | 72.41 | 69.78 | 91.30 | 77.36 | 74.41$_{\pm 0.17}$ | 0.28 | 94 |
| +FedHEAL [25] | 86.50 | 53.18 | 74.33 | 72.42 | 90.65 | 80.00 | 76.18$_{\pm 0.30}$ | 2.05 | 53 |
| +DynaFed [27] | 86.13 | 50.31 | 69.22 | 85.90 | 87.16 | 81.40 | 76.68$_{\pm 1.27}$ | 2.55 | 100 |
| +FedFTG [19] | 84.31 | 50.00 | 71.20 | 80.87 | 87.69 | 74.69 | 74.79$_{\pm 0.32}$ | 0.66 | 114 |
| +FGL [20] | 91.09 | 55.56 | 79.71 | 79.07 | 92.21 | 84.80 | 80.41$_{\pm 0.39}$ | 6.28 | **5** |
| **+Flick** | 92.11 | 63.90 | 83.88 | 91.40 | 92.91 | 90.34 | **85.76**$_{\pm 0.07}$ | **11.63** | 11 |
| *FedProx* [9] | 85.32 | 52.59 | 72.99 | 83.37 | 88.15 | 80.76 | 77.20$_{\pm 0.60}$ | - | 144 |
| +FedBN [26] | 83.48 | 52.73 | 72.34 | 66.63 | 91.49 | 77.66 | 74.06$_{\pm 0.02}$ | -3.14 | - |
| +FedHEAL [25] | 87.36 | 54.84 | 74.88 | 75.91 | 92.33 | 81.86 | 77.86$_{\pm 0.12}$ | 0.66 | 58 |
| +DynaFed [27] | 86.94 | 53.94 | 73.18 | 82.60 | 87.56 | 81.79 | 77.67$_{\pm 0.11}$ | 0.47 | 115 |
| +FedFTG [19] | 86.82 | 51.79 | 73.86 | 86.07 | 89.72 | 81.45 | 78.28$_{\pm 0.30}$ | 1.08 | 114 |
| +FGL [20] | 91.40 | 54.98 | 80.98 | 81.00 | 92.79 | 85.03 | 81.03$_{\pm 0.48}$ | 3.83 | **14** |
| **+Flick** | 92.81 | 64.94 | 84.94 | 91.60 | 93.07 | 89.66 | **86.17**$_{\pm 0.05}$ | **8.97** | 21 |
| *FedDyn* [49] | 88.06 | 55.39 | 77.08 | 75.60 | 90.32 | 81.03 | 77.91$_{\pm 0.22}$ | - | 34 |
| +FedBN [26] | 87.22 | 54.79 | 78.82 | 77.90 | 89.72 | 80.48 | 78.16$_{\pm 0.33}$ | 0.25 | 45 |
| +FedHEAL [25] | 86.51 | 56.97 | 74.66 | 75.03 | 91.89 | 82.68 | 77.96$_{\pm 0.21}$ | 0.05 | 93 |
| +DynaFed [27] | 84.41 | 53.53 | 74.11 | 81.73 | 89.46 | 81.03 | 77.38$_{\pm 0.21}$ | -0.53 | - |
| +FedFTG [19] | 86.13 | 54.05 | 74.50 | 82.20 | 88.80 | 81.93 | 77.94$_{\pm 0.59}$ | 0.03 | 31 |
| +FGL [20] | 87.35 | 54.67 | 76.55 | 80.17 | 89.99 | 82.76 | 78.58$_{\pm 0.13}$ | 0.67 | 31 |
| **+Flick** | 89.68 | 60.37 | 83.09 | 87.87 | 91.14 | 87.03 | **83.20**$_{\pm 0.13}$ | **5.29** | **26** |
| *FedNAR* [16] | 89.88 | 56.85 | 75.36 | 86.10 | 91.69 | 81.93 | 80.30$_{\pm 0.40}$ | - | 150 |
| +FedBN [26] | 85.50 | 58.22 | 76.50 | 78.90 | 93.33 | 81.17 | 78.94$_{\pm 0.14}$ | -1.36 | - |
| +FedHEAL [25] | 88.43 | 57.05 | 77.37 | 77.87 | 93.87 | 83.69 | 79.71$_{\pm 0.02}$ | -0.59 | - |
| +DynaFed [27] | 89.77 | 56.08 | 77.16 | 83.63 | 92.60 | 84.38 | 80.60$_{\pm 0.21}$ | 0.30 | 150 |
| +FedFTG [19] | 90.72 | 53.10 | 74.63 | 84.50 | 94.71 | 84.41 | 80.34$_{\pm 0.10}$ | 0.03 | 114 |
| +FGL [20] | 90.38 | 57.47 | 81.11 | 83.83 | 92.65 | 85.24 | 81.78$_{\pm 0.07}$ | 1.48 | 79 |
| **+Flick** | 91.30 | 64.73 | 84.02 | 90.40 | 93.11 | 89.10 | **85.44**$_{\pm 0.17}$ | **5.14** | **36** |

# D  Overheads

Compared to the model-driven methods for heterogeneous FL, the data-driven Flick introduces additional overheads to the clients, including extra training and captioning latency. However, the

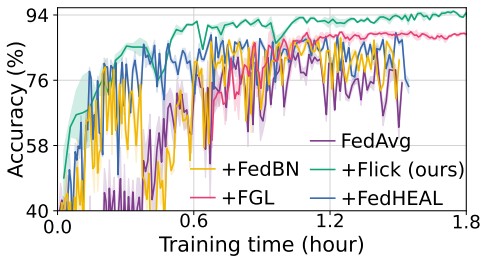

(a) Time-to-Accuracy performance

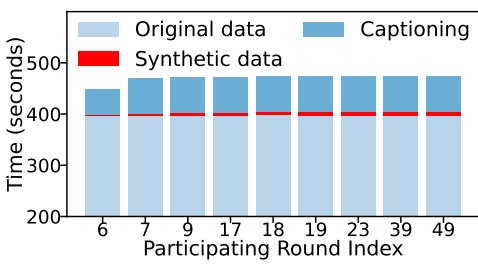

(b) Local latency over rounds

Figure 14: Evaluation on the temporal scale: **(a)** Wall-clock training time of Flick and its model-driven counterparts on the PACS dataset, running on an NVIDIA A40 GPU; **(b)** Wall-clock local latency of a specific client, running on an NVIDIA Jetson AGX Orin, across participating rounds, including training on both original and synthetic data points, as well as local captioning.

Table 9: Resource usage comparison between model-driven FedProx and data-driven Flick across three benchmarks. We report per-round and accumulated-to-target (with target defined as the best performance of FedProx, highlighted in gray) resource consumption using four metrics: *Peak GPU Memory (PGM)*, *GPU Memory Hours (GMH)*, *Energy (E)*, and *Running Time (RT)*. For GMH and E, we provide the dynamic range observed across the communication rounds. **Lowest** in bold.

| Metrics | PACS | | Office-Caltech | | DomainNet | |
|---|---|---|---|---|---|---|
| | FedProx | Flick | FedProx | Flick | FedProx | Flick |
| PGM (GB) | 8.33 | 14.53 | 4.77 | 10.50 | 13.74 | 20.47 |
| GMH (GB·hour) | [0.049, 0.064] | [0.152, 0.353] | [0.016, 0.022] | [0.084, 0.376] | [0.529, 0.681] | [1.407, 2.110] |
| E×10$^{-4}$ (kWh) | [0.89, 80.62] | [5.64, 164.60] | [1.81, 52.47] | [1.89, 90.90] | [8.23, 273.60] | [8.06, 1207.00] |
| RT (hour) | 0.96 | **0.52** | 0.55 | **0.50** | 6.82 | **2.11** |
| GMH (GB·hour) | 7.91 | **5.92** | **2.63** | 5.25 | 91.79 | **34.61** |
| E (kWh) | 0.149 | **0.075** | 0.063 | **0.050** | 1.329 | **0.444** |

performance gain from Flick remains significant: it facilitates the model convergence, as reflected in both reduced round-to-accuracy (#Round) and training time, while also improving the top-1 accuracy of the global model in the long run.

Figure 14(a) shows the learning curves of Flick and its counterparts, integrated into the baseline method FedAvg, with time-to-accuracy performance evaluated on an NVIDIA A40 GPU. The faster convergence offered by Flick significantly counteracts the negative impact from additional overheads.

We also trace local latency variation over communication rounds by measuring the time consumption of a single client running on an NVIDIA Jetson AGX Orin, as shown in Figure 14(b). We break down the additional overheads into training on synthetic data and captioning the representative samples, which employs the "Salesforce/blip-image-captioning-large" with ViT and BERT as image and text encoders. It is obvious that the local overheads of Flick stabilize over time and constitute only a small fraction of the overall local latency as training proceeds. In contrast, FGL – the best-performing counterpart – incurs substantial delay in the first communication round due to the extensive preparation of synthetic data, leading to a longer time to reach the target accuracy.

We further investigate storage and GPU memory requirements across three VLP-based captioning models: BLIP-Large requires 1.88 GB of storage and 3.19 GB of GPU memory; BLIP-Base requires 0.97 GB of storage and 2.30 GB of GPU memory; ViT-GPT2 requires 0.96 GB of storage and 2.48 GB of GPU memory. Moreover, Flick shows consistent performance gains across these three image captioning models as shown in Figure 11. Such flexibility enables clients to choose models that are best suited to their available computational budgets.

Additionally, we compare the resource usage of data-driven Flick with the model-driven baseline FedProx from three aspects: 1) peak GPU memory (GB) during training, 2) GPU memory hours (GB·hour), akin to GPU hours but weighted by runtime memory usage, and 3) energy consumption (kilowatts·hour). We collect these statistics on an NVIDIA A40 GPU for both FedProx and Flick across the three benchmarks. The Table 9 reports the per-round resource usage and the accumulated usage required to reach the target accuracy (i.e., best accuracy of FedProx). Since GPU memory

hours and energy consumption vary across communication rounds, we provide both the dynamic range and the average value over all rounds. For Flick, GPU memory hours and energy consumption show a wider dynamic range because the overhead from key components changes across rounds. Specifically, Flick relies 'heavily' on LLM and LDM for synthetic data generation during the early training stage, which takes the majority of the overall cost. As training progresses, both GPU memory hours and energy consumption per round in Flick decrease. In contrast, the variation in these two per-round measurements for FedProx is mainly due to the randomness of participating clients in each round, where heterogeneous data silos introduce varying local training overhead.

From the results, we observe that for the Office-Caltech dataset, Flick requires more GPU memory hours than FedProx to reach the target accuracy. This is because the additional GPU memory for LLM and LDM far exceeds that required for local training, thereby counteracting the resource savings from faster convergence (i.e., fewer communication rounds). However, Flick still offers shorter training latency to the target and achieves a higher best accuracy than FedProx. For more complex tasks, such as the DomainNet benchmark with large-scale clients and data, Flick outperforms conventional FL across all metrics: it reaches the same performance as counterpart FL frameworks with lower training latency, GPU memory usage, and energy cost; furthermore, it achieves better accuracy in the long run. To conclude, although Flick introduces additional overhead, primarily on the server, it enables faster convergence of the global model compared to FedProx. In most cases, Flick further reduces resource consumption to achieve the same target accuracy, particularly in challenging settings with large-scale clients and datasets. Furthermore, with continued training, Flick offers substantial accuracy gains; for example, on the PACS dataset, Flick achieves 11.43% higher accuracy than FedProx after the same number of communication rounds.

# E  Privacy Analysis

Flick can ensure the $\epsilon$-DP guarantee by adding noise sampled from the Laplace distribution to the token sequences, with the scale parameter in the Laplace mechanism set to $1/\epsilon$. It provides stronger privacy protection as $\epsilon$ decreases. Table 10 shows the trade-off between system utility (i.e., Flick performance) and privacy level, with the $\epsilon$ ranging from 0.05 to 0.5. Note that the performance gradually degrades towards the Flick *w/o Local Summary* variant as the $\epsilon$ decreases. Even under strict privacy constraints, Flick maintains promising model performance.

Table 10: The performance of Flick across different $\epsilon$ values.

| $\epsilon$ | PACS | | | Office-Caltech | | |
|---|---|---|---|---|---|---|
| | AVG ↑ | $\Delta$ ↑ | #Round ↓ | AVG ↑ | $\Delta$ ↑ | #Round ↓ |
| $\epsilon = 0.5$ | 94.32±0.21 | 11.26 | 11 | 78.24±0.15 | 7.40 | 38 |
| $\epsilon = 0.1$ | 93.78±0.24 | 10.72 | 25 | 77.61±0.33 | 6.77 | 43 |
| $\epsilon = 0.05$ | 92.99±0.31 | 9.93 | 31 | 77.05±0.32 | 6.21 | 43 |

Table 11: Privacy assessment of local summary phase in Flick. The server utilizes five image generation models to reconstruct the data from collected local summaries. The similarity between original raw local data and reconstructed data is measured by three widely used metrics, with reference ranges provided to indicate the threshold below/beyond which pair-wise images can be considered dissimilar. *PSNR [54]: peak signal-to-noise ratio; SSIM [55]: structural similarity index measure; LPIPS [56]: learned perceptual image patch similarity.*

| Metrics | dall-e-2 | dall-e-3 | v1.5 | v1.4 | xl-base-1 |
|---|---|---|---|---|---|
| PSNR ($< 20$) | 7.77 | 8.11 | 7.79 | 7.80 | 9.37 |
| SSIM ($< 0.3$) | 0.20 | 0.16 | 0.17 | 0.14 | 0.20 |
| LPIPS ($> 0.6$) | 0.72 | 0.74 | 0.78 | 0.77 | 0.72 |

Besides, we assess the potential information leakage in local summaries from a different angle: the difficulty for the server to infer raw data from collected token sequences. To this end, the central server feeds the local summaries (in the textual format) to a text-to-image model, and we evaluate five different models. Table 11 reports the reconstruction quality based on three widely used metrics–peak signal-to-noise ratio (PSNR) [54], structural similarity index measure (SSIM) [55], and learned

perceptual image patch similarity (LPIPS) [56]–which measure the similarity between reconstructed images generated by server-held models and the original raw local images. For PSNR and SSIM, higher values indicate greater similarity, whereas for LPIPS, lower values indicate higher similarity. The results across all five image generators show that reconstructed images remain significantly dissimilar to the original raw data across three metrics, demonstrating that uploading local summaries in Flick presents a low privacy risk.

# F Practical Impact

Although versatile foundation models are rapidly advancing and becoming increasingly accessible, the design of Flick remains centered on training task-specific lightweight models – a crucial requirement for edge deployment under privacy constraints and limited computational resources. To articulate the practical impact of Flick, we recognize it is important to clarify the following two questions:

*Given the rapid advancement of versatile foundation models, what role do conventional models (e.g., ResNet) still play?*

and the follow-up:

*How can such powerful foundation models be leveraged to benefit the training of lightweight, task-specific models, especially in a privacy-preserving federation way?*

For the first question, despite foundation models offering broad generalization capabilities across a wide range of tasks, directly deploying them in real-world applications, especially on resource-constrained edge devices, is still challenging (exceeding resource budgets) and is overkill (leading to unacceptable latency) for specific tasks. Alternatively, relying on remote cloud-based inference via APIs introduces significant privacy risks, as it requires users to upload private or sensitive raw data to third-party services. Together, these limitations indicate that both the versatile LLMs and lightweight models offer practical values depending on deployment constraints such as resource budgets, latency tolerance, and privacy preferences. Consequently, there remains a strong and persistent demand for lightweight task-specific models that balance efficiency and effectiveness in deployment.

Table 12: Performance comparison between Qwen-series MLLMs and a lightweight ResNet-18 on the DomainNet dataset. Note that ResNet-18 is trained under a federated setting with both domain shift and label skew across silos. AVG denotes the average accuracy calculated across all domains.

| Methods | DomainNet | | | | | | |
|---|---|---|---|---|---|---|---|
| | Clipart | Infograph | Painting | Quickdraw | Real | Sketch | AVG |
| Qwen2-VL-7b [57] | 96.97 | 91.08 | 93.53 | 63.13 | 97.05 | 98.07 | 89.97 |
| Qwen2-VL-2b [57] | 97.37 | 90.66 | 92.21 | 62.07 | 96.78 | 97.79 | 89.48 |
| Qwen2.5-VL-7b [58] | 97.77 | 92.32 | 94.32 | 62.73 | 97.31 | 98.21 | 90.44 |
| Qwen2.5-VL-3b [58] | 96.36 | 88.59 | 92.73 | 57.27 | 97.18 | 96.55 | 88.11 |
| FGL [20] | 91.09 | 55.56 | 79.71 | 79.07 | 92.21 | 84.80 | 80.41 |
| **Flick** | 92.11 | 63.90 | 83.88 | 91.40 | 92.91 | 90.34 | 85.76 |

To support our claims, we investigate the runtime performance of Qwen-series [57, 58] Multimodal Large Language Models (MLLMs) on the DomainNet benchmark, and report the inference latency on an NVIDIA Jetson AGX Orin. Under GPU-mode, Qwen2.5-VL-3B takes 7.05 seconds per sample, while the lighter Qwen2-VL-2B still requires 5.94 seconds. The latency is severely exacerbated under CPU-only mode: 239.43 seconds for the Qwen2.5-VL-3B and 169.93 seconds for the Qwen2-VL-2B. We emphasize that the Jetson AGX Orin is already considered powerful hardware for edge computing. In contrast, ResNet-18 requires only 4.73 seconds per sample under the same CPU-only configuration. When using GPU acceleration, the inference time is further reduced to 0.4 seconds per sample.

We further compare the classification performance of Qwen-series MLLMs with a lightweight ResNet-18 trained under Flick framework on the DomainNet benchmark, as shown in Table 12. We emphasize two points for fairly assessing the ResNet-18 performance and Flick contributions: 1) training under heterogeneous FL settings inherently degrades model performance compared to centralized training; and 2) our data-driven Flick addresses such challenges more effectively than the similar LLM-based FGL by efficiently incorporating commonsense knowledge from LLMs. The results show that Flick achieves competitive global accuracy on DomainNet while maintaining edge

practicality. We also note an interesting observation from the DomainNet results, particularly on the two challenging domains, 'Infograph' and 'Quickdraw': all four MLLMs perform relatively well on the 'Infograph' domain, while showing generally inferior performance on the 'Quickdraw' domain. As described in DomainNet [45], 'Quickdraw' images are collected from Google's 'Quick, Draw!' game, where the drawings are very simple, abstract line sketches; 'Infograph' images are typically informational graphics such as posters, diagrams, and other structured visual layouts. This distinct performance gap between the two domains suggests that: 1) MLLMs inherently have a strong understanding of complex icons, symbols, and text arranged in a structured format; and 2) they still have limitations in handling certain tasks. This observation underscores the value of FL frameworks, which allow privacy-preserving model training for such domain-specific tasks using task-relevant, decentralized data.

By answering the follow-up questions, we aim to highlight the contribution of our data-driven Flick in improving the performance under heterogeneous FL by effectively leveraging powerful foundation models. In the FL context, conventional approaches focus on optimizing training over distributed data silos, which inherently limits the upper bound of accessible knowledge. This performance ceiling can be lifted by distilling external task-related knowledge embedded in foundation models, which is actually our target in Flick.

# G    Data Generation Details

In Section 3.3, we have introduced the data generation process in Flick. In this section, we provide some more details – specifically for the LLM usage and data retrieval – by showing the intermediate results.

The data generation phase bootstraps the server to obtain the class-wise decision matrix $\mathbf{D}$. After that, an LLM is employed to generate text prompts for each class. Figure 15 provides a detailed case example to supplement Figure 3 in the paper, illustrating how the LLM produces $G = 5$ text prompts for the class *dog*, using a collection of local captions $\mathcal{T}$ as reference. Recall that, for privacy considerations, our design in the local summary phase only requires participating clients to report captions of representative local samples while omitting class information. Consequently, the LLM has to first identify and select class-related captions from the pool $\mathcal{T}$. From the LLM output in Figure 15, it is evident that the LLM can effectively select captions relevant to the class *dog*, extending beyond simple keyword matching. For instance, it also identifies captions containing terms like

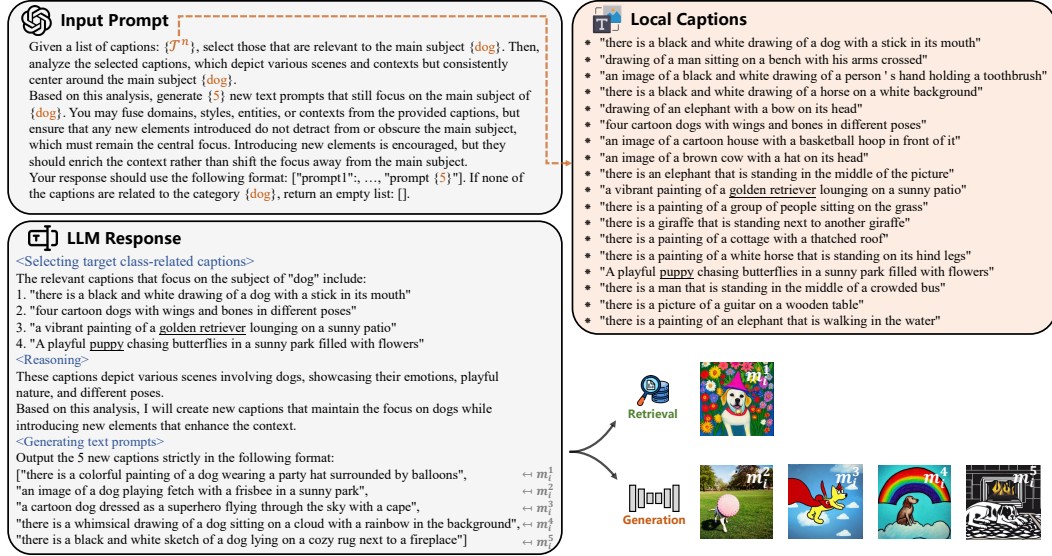

Figure 15: Detailed illustration of LLM-based text prompt generation: given cross-client-specific knowledge $\mathcal{T}^n$, the central server obtains the text prompts for the class *dog* in the PACS dataset. The underlined words are recognized by LLM as related to class *dog*.

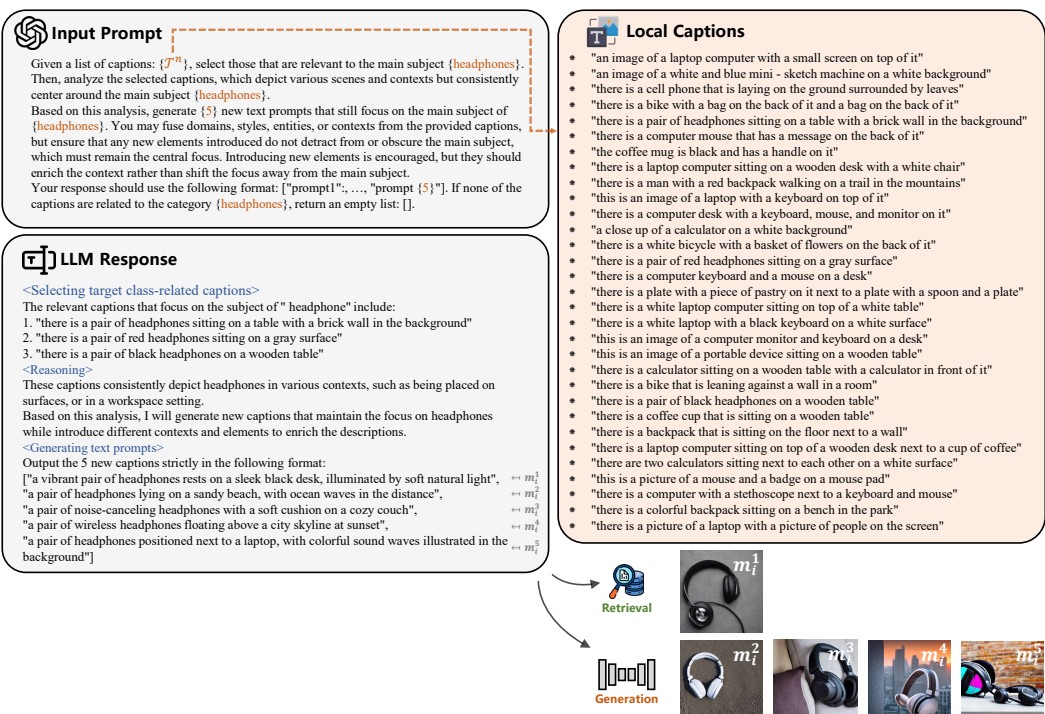

Figure 16: Detailed illustration of the LLM-based text prompt generation for the class *headphones* in the Office-Caltech dataset.

"golden retriever" or "puppy". During the reasoning phase, the LLM analyzes the selected captions and recombines extracted features (i.e., cross-client-specific knowledge) with new elements (i.e., commonsense knowledge embedded in the LLM) to generate text prompts for the subsequent text-to-image model. Besides, Figure 16 illustrates the text prompts generation for the class *headphones* in the Office-Caltech benchmark where the LLM showcases similar capabilities in extracting class-related knowledge and instilling commonsense knowledge.

To reduce the data generation overheads, the server in our Flick does not directly feed all text prompts into a generative model for new data points. Instead, it first retrieves historically generated samples from the server-held data pool $\mathcal{G}_s$. The qualified historical samples are retrieved based on the pairwise similarity between the historical text prompt $m_s^p \in \mathcal{M}_s$ and the generated text prompt $m_i^q \in \mathcal{M}_i$. Specifically, each prompt pair $(m_s^p, m_i^q)$ is transformed into SBERT embeddings used for calculating cosine similarity. In Figure 17, we provide more examples of the data retrieval step across two benchmarks and their respective four domains. For detailed illustrations, Figure 17 presents the text prompt pairs $(m_s^p, m_i^q)$ and their SBERT similarity scores, as well as the sample corresponding to the historical text prompt $m_s^p$ that the server retrieves.

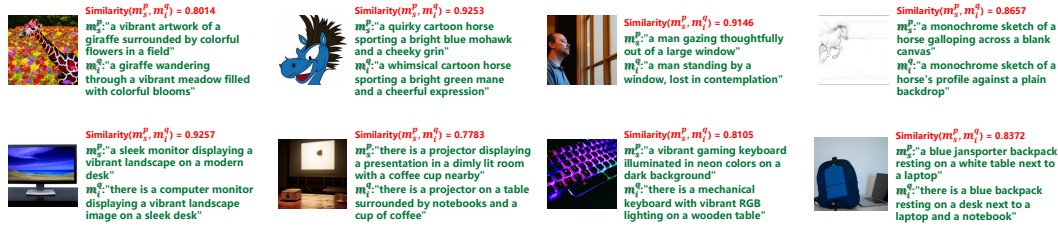

Figure 17: Illustration of the retrieved samples in two datasets: PACS (*top*) and Office-Caltech (*bottom*). In each example, $m_s^p$ and $m_i^q$ represent historical text prompt and generated text prompt, respectively. In Flick, the server retrieves samples from the data pool for which the corresponding text prompts show high similarity to the generated text prompts.

