# OpenReview forum: "Flick: Empowering Federated Learning with Commonsense Knowledge"
_NeurIPS.cc/2025/Conference — NeurIPS 2025 poster_

### Official Review · Reviewer_FvW9 · 2025-06-22

**Clarity:** 3
**Significance:** 2
**Originality:** 2
**Rating:** 4
**Confidence:** 4

**Summary:**

To address the label skew and domain shift of data silos across clients in FL, this paper proposes Flick, a generative data augmentation framework for heterogeneous FL, which introduces three additional phases into the FL workflow, including local summary, data generation, and generated data usage. On the one hand, clients select representative samples from their local data and use an image-to-text model as a caption generator to transform samples into low-sensitivity tokens, extracting client-specific knowledge from local data, which is then offloaded to the server. On the other hand, the server uses an out-of-the-box LLM to analyze local summaries and instill inherent commonsense knowledge into generated text prompts, which direct a text-to-image model in producing synthetic data, enabling global model fine-tuning and local data compensation to gradually align the label and feature distributions across clients. Experiments on three multi-domain image classification datasets (PACS, Office-Caltech, and DomainNet) demonstrate the effectiveness of the proposed method.

**Questions:**

N/A

**Ethical Concerns:**

["NO or VERY MINOR ethics concerns only"]

**Final Justification:**

After reading the authors' rebuttal,  most of my misunderstandings and concerns have been addressed, so I tend to keep the original score and maintain a positive score.

**Limitations:**

The authors partially addressed the limitations and potential negative societal impact of their work.

**Quality:**

3

**Strengths And Weaknesses:**

**Reasons to Accept:**

1.	The method is innovatively designed, skillfully integrating image-to-text models, LLMs, sentence embedding models, and text-to-image models with synthetic data generation.
2.	The synthetic data are used not only for fine-tuning the global model but also for local data compensation at the clients, creating a dual-benefit effect.
3.	The experiment is thorough, and the implementation details are clear.

**Reasons to Reject:**

1.	Although synthetic data is used to improve model performance, the paper lacks quantitative evaluations of image quality (e.g., FID, IS, CLIP score), making it difficult to establish a direct link between data quality and performance gains.
2.	The applicability of Flick to non-vision tasks (e.g., text classification, speech, time series) or more complex structured tasks (e.g., object detection, segmentation) is not discussed.
3.	While Flick aims to alleviate data heterogeneity, its mechanism—relying on "client feedback per round → server-generated data → client-side compensation"—may introduce new forms of client imbalance (e.g., favoring clients with stronger descriptive capabilities), due to variations in the amount of feedback, description quality, and generated data quality. No fairness or consistency analysis is provided in this regard.
4.	Token length and communication cost statistics are not reported.
5.	Compared with traditional federated learning methods, the proposed method is very complex, introducing four additional large-scale models and multiple hyperparameters, including an image-to-text model (i.e., Salesforece/blip-image-captioning-large) for image captioning of local data, a large language model (i.e., gpt-4o-mini) for text prompt generation, a sentence embedding model (i.e., Sentence-BERT) for similarity calculation between generated text prompts and pooled text prompts, and a text-to-image model (i.e., sd-legacy/stable-diffusion-v1-5) to synthesize new data points from low-similarity text prompts. Therefore, it brings concerns on training burden and optimization difficulty.
6.	The authors did not provide any code for review, so the reproducibility of this paper is open to question.

---

> ### Author Rebuttal · Authors · 2025-07-31
>
> We thank the reviewer for the insightful comments. Below are our responses.
>
> `[1/6] The paper lacks quantitative evaluations of image quality (e.g., FID, IS, CLIP score), making it difficult to establish a direct link between data quality and performance gains.`
>
> The data generation in Flick follows two steps: the server first obtains a set of text prompts from LLM, then feeds them into a text-to-image model (i.e., LDM) for synthetic images. We use an out-of-the-box image generation model (e.g., stable-diffusion-v1-5 used in the experiments) without fine-tuning. Therefore, metrics such as FID, IS, or CLIP score, which primarily evaluate the pretrained generator, are not the focus of our evaluation. Instead, as shown in Figure 12 in Appendix C, Flick consistently improves performance across different image generators, highlighting its resilience to the image generative backend.
>
> We clarify that the performance gain mainly lies in the generation of informative, task-relevant text prompts, which is the core contribution of this work. To this end, we propose a prompt template that instructs LLM in extracting information from the collected cross-client-specific knowledge while instilling inherent commonsense knowledge to produce informative text prompts for each class. As shown in Table 2, integrating cross-client-specific knowledge leads to significant gains over local generation. Further, the ablation results in Table 3 confirm that the inclusion of commonsense knowledge improves text prompt effectiveness.
>
> `[2/6] The applicability of Flick to non-vision tasks (e.g., text classification, speech, time series) or more complex structured tasks (e.g., object detection, segmentation) is not discussed.`
>
> We clarify that the Flick can be extended to other vision tasks with minimal modification, particularly in how to obtain the label $y_s$ of a generated data point $(x_ s, y_ s)$. In the image classification task investigated in this paper, the label $y_ s$ refers to the image class, which is known a priori to the server. When adapting Flick to more complex tasks such as object detection, $y_ s$ would include both the class label and the bounding box of the object. To this end, the server can leverage an automated annotator, such as SAM [1], to generate structured bounding boxes for synthetic images.
>
> We further highlight that the core contribution of Flick lies in its data-driven pipeline, which integrates low-sensitive client-specific knowledge distillation, LLM-based commonsense knowledge instillation, and dual-use of generated samples to address data heterogeneity in federated learning. This pipeline is modality-agnostic and not limited to vision tasks. Nevertheless, we acknowledge two key challenges when applying Flick to non-vision modalities such as text, speech, or time series: 1) identifying suitable methods for converting raw data into meaningful yet low-sensitive text representations (akin to image captioning); and 2) finding the generative models with the capacity of producing realistic synthetic data from text prompts. We leave this as a promising direction for future work.
>
> [1] Alexander Kirillov, et al. "Segment anything". ICCV, 2023.
>
> `[3/6] While Flick aims to alleviate data heterogeneity, its mechanism—relying on "client feedback per round → server-generated data → client-side compensation"—may introduce new forms of client imbalance (e.g., favoring clients with stronger descriptive capabilities), due to variations in the amount of feedback, description quality, and generated data quality. No fairness or consistency analysis is provided in this regard.`
>
> We highlight two essential fairness-enhancing features built into Flick. First, all clients have an equal opportunity to share their client-specific knowledge with the server. In each communication round, the server randomly selects a set of clients, and each participating client is allowed to report the same number of local summaries --  specifically, one representative local sample per class for image captioning. Second, the quality of generated data used for local compensation is consistent across clients, as the server centrally employs a text-to-image model (i.e., LDM) to synthesize data. This generation process is decoupled from individual client differences and thus not influenced by client diversity.
>
> While variations in description quality, mainly arising from different local captioning models across clients, may still introduce fairness concerns, we mitigate this by enforcing a fixed output length (default: 30 tokens) across all captioning models. This constraint helps limit variations in description quality. Moreover, designing a fairness-aware prompt template could further alleviate the negative impact of local summary disparities on LLM analysis, which we leave as a promising direction for future work.
>
> `[4/6] Token length and communication cost statistics are not reported.`
>
> For the local summary phase, each client uses a VLP model to selectively caption one representative sample per class. We set the maximum token length of each image caption to 30 tokens, balancing the informativeness of the content with the risk of privacy leakage.
>
> We break down the communication overhead in Flick into three parts: uploading local summaries, downloading synthetic samples for local data compensation, and exchanging model parameters, where the first two components represent additional overhead compared to FedAvg. Taking the PACS experiment in Table 1 as an example, each client reports local summaries with a minimal cost ranging from $1.01kb$ to $1.48kb$ per round, which is negligible compared to the $42.72Mb$ for exchanging model parameters. For a fair comparison, we report the overall communication overhead when Flick and FedAvg reach the target accuracy (defined as the best accuracy of FedAvg). Flick requires only 11 communication rounds with the server sending 510 synthetic images to clients. Thus, the total communication cost of Flick is: $42.72 (Mb/round) \times 11 (round) \times 2 + 0.14 (Mb/sample) \times 510 (sample)=0.99Gb$ where the factor 2 refers to clients offloading the local updates and downloading the global model. For FedAvg, it takes 142 rounds to reach the same target accuracy, and the total communication cost is $42.72 (Mb/round) \times 142 (round) \times 2 =11.85Gb$. Moreover, after 150 rounds, Flick further improves the global model accuracy by 11.43% over FedAvg, by compensating 2,405 synthetic samples at an additional cost of only $0.33Gb$. It indicates that the performance gain by Flick justifies the additional costs.
>
> `[5/6] Compared with traditional federated learning methods, the proposed method is very complex, introducing four additional large-scale models and multiple hyperparameters. Therefore, it brings concerns on training burden and optimization difficulty.`
>
> We clarify that the image captioning model (i.e., VLP model), LLM, SBERT embedding model, and image generation model (i.e., LDM) are all out-of-the-box models without any fine-tuning or prompt-tuning when applied in our Flick framework. As a result, they introduce zero training burden to Flick. Moreover, the inference overhead introduced by these models is controllable and flexible: 1) three of the four models (excluding the VLP model) are running on the server, which typically has rich computing resources; 2) the performance gain from Flick is consistent to the choice of image captioning model, LLM, and image generation model, as evidenced by the results in Figures 10, 11, and 12 in Appendix C, respectively. This flexibility allows both clients and the server to select models that are best suited to their available computational budgets.
>
> We also investigate the sensitivity of Flick to the key hyperparameters: text similarity $T_ s$, validation accuracy $T_ v$, and data pool size $\lvert \mathcal{G}_ {s}\lvert$ in Figure 9, and data generation budget $G$ in Table 5 of Appendix C. The extensive experimental results show that Flick generalizes well without requiring careful hyperparameter tuning.
>
> `[6/6] The authors did not provide any code for review, so the reproducibility of this paper is open to question.`
>
> We appreciate the reviewer for pointing it out. Due to the policy prohibiting external links in the rebuttal, we are unable to share the code at this stage. We are committed to releasing the code once anonymity is no longer required.

---

> > ### Comment · Reviewer_FvW9 · 2025-08-07
> >
> > Thank you for your reply. The authors' rebuttal has addressed most of my misunderstandings and concerns, so I tend to keep the original score and maintain a positive score.

---

> > > ### Author Response · Authors · 2025-08-07
> > >
> > > Dear Reviewer FvW9,
> > >
> > > Thank you for taking the time to review our rebuttal. We are glad that our responses have addressed your concerns and clarified the points of misunderstanding. We would be happy to provide any additional follow-up clarifications to questions you may have.
> > >
> > > Best,
> > > Authors

---

### Official Review · Reviewer_28YM · 2025-06-23

**Clarity:** 3
**Significance:** 2
**Originality:** 3
**Rating:** 4
**Confidence:** 4

**Summary:**

This paper proposes **Flick**, a novel data generation framework for heterogeneous Federated Learning with Commonsense Knowledge from Large Language Models.
Within Flick, the client performs the local data summary to capture client-specific knowledge in textual form.
The server then generates new data samples by analyzing the collected captions using an LLM guided by a designed prompt. Generated samples are stored in a server-maintained data pool, which supports both global model fine-tuning and local data compensation.
Flick achieves superior performance, as demonstrated by experiments on two datasets, PACS and Office-Caltech.

**Questions:**

**Questions**:
1. How many data samples are used in Table 1? This information is difficult to find, even after lookig through both the main text and the appendix. Specifically, please clarify the number of:

    Original data samples, prompts generated (is G = 10?), generated images, and sampled images for training.
***
**Suggestions**:
1. Make clear which LLM/Diffusion model you use in "Datasets and Models".
***

**Ethical Concerns:**

["NO or VERY MINOR ethics concerns only"]

**Final Justification:**

At least half of my concerns regarding the performance-efficiency trade-off have been addressed.
While I still hold some reservations about the practical value of Flick, I believe—following the principle of "focusing on merits"—that its inherent limitations should not be too much criticized by me.
Therefore, I am now leaning toward acceptance in respond to the authors' hard efforts.

**Limitations:**

I acknowledge that the authors have made a genuine effort to discuss the limitations of this paper, which primarily relate to the additional cost overhead and potential privacy risks.
However, I remain concerned about the severity of these limitations and their impact on the practical applicability of the proposed method.

**Quality:**

3

**Strengths And Weaknesses:**

**Strengths**:
1. Extensive experiments with 9 baselines on two datasets.
2. Clear structure and visualization.
3. The loss-based sample selection method, aimed at minimizing privacy risks, is both critical and effective.
4. Additional experiments in the Appendix to validate the choice of hyper parameters.

***
**Weaknesses**:
1. I greatly appreciate the authors' efforts in presenting Flick. However, the method appears to rely on a major, and arguably unreliable, assumption: *leveraging powerful LLMs and diffusion models to assist in training ResNet models for relatively simple image classification tasks* and *via federated learning*. This approach seems misaligned with current community practices and **lacks clear practical applications**. To draw an analogy, it feels like hiring an astronaut to ride a bicycle.
2. Building on the first point, although Appendix D provides some analysis, the proposed approach introduces substantial additional computational overhead—not all of which is adequately addressed in Appendix D. Some of these additional costs are non-negligble:
    * the deployment cost of diffusion models which is significantly higher compared to that of standard ResNet models;
    * the API fees associated with using GPT models;
    * the inference time required for both LLMs and diffusion models to generate synthetic data;
    * etc.

    Have you measured the actual time required to generate synthetic data in Figure 14? Based on my estimates, the generation time probably far exceeds 500 seconds.

    If the authors can provide a detailed breakdown of the time and resource consumption involved in the full Flick process—especially in comparison with traditional federated learning methods such as FedProx—and convincingly argue that the additional costs are justified by performance gains, I would be happy to raise my score.

3. Lack of comparison with other straightforward approaches.
    * If the server has already acquired summaries of the client data and can generate synthetic data accordingly, why not simply fine-tune a global model on all synthetic data directly at the server side? This would reduce communication to just a single round, which, according to Table 1, Table 2, and Figure 6, yields the best statistical performance.
    * If powerful LLMs or diffusion models are assumed to be available, why not directly leverage them for the task? It is necesary to either compare the proposed approach with models such as GPT-4o or Qwen2-VL on PACS, or justify why these models are not suitable in the given context.
4. Lack of experimental comparisons under standard IID settings, e.g. using datasets like CIFAR.
***

---

> ### Author Rebuttal · Authors · 2025-07-31
>
> We thank the reviewer for the insightful comments. Below are our responses.
>
> `[1/5] Lack practical applications, and why not leverage powerful LLMs for the task?`
>
> Despite the increasing availability of versatile foundation models, we emphasize that real-world deployment, especially inference at the edge, still requires task-specific lightweight models due to privacy constraints and limited resources. This also necessitates privacy-preserving training on data that remains distributed across users, thereby leading to the emergence of FL. On one hand, even if a strong pretrained model (e.g., ResNet-18 trained on ImageNet) is available, it still requires refinement using task-relevant, decentralized data residing on edge devices. On the other hand, while foundation models such as GPT-4o or Qwen2-VL may achieve superior performance on tasks like PACS classification, they are typically too large to run efficiently on edge devices. Relying on remote cloud-based inference via APIs introduces significant privacy risks, as it requires users to upload private or sensitive raw data to third-party services.
>
> `[2/5] Authors can provide a breakdown of time and resource consumption involved in the Flick process; API fees associated with using GPT.`
>
> Thanks for the suggestion. Below, we provide a detailed breakdown of the additional computational overhead introduced by Flick, which varies across rounds.  For example, we report results from an experiment running on an NVIDIA A40 GPU, using the FedProx baseline on PACS dataset. The table below shows the wall-clock time (in seconds) for six components: extra local training, local summary,  LLM inference, sample retrieval, data generation, and global model fine-tuning. We observe that the server-side computational overhead (i.e., the last four components) takes the majority of the overall cost. However, server-side overhead decreases over time, as Flick no longer heavily requires data generation after 20 rounds. We also clarify that the data generation time using the LDM is approximately **1** second per sample, which is lower than the reviewer's estimation. This latency can be further reduced with a more powerful server.
>
> |Round|Extra Local Training|Captioning|LLM|Retrieval|Generation|Global Finetuning|
> |-|-|-|-|-|-|-|
> |1|0.00░░░░░|3.18▓▓▓▓▓|15.90▓▓▓▓▓░|0.00░░░░░░|31.18▓▓▓▓▓▓░░░░░░|0.35▓░░░░|
> |3|0.84▓▓░░░|3.05▓▓▓▓▓|18.55▓▓▓▓▓▓|0.97▓░░░░░|51.88▓▓▓▓▓▓▓▓▓▓░░|0.42▓░░░░|
> |5|1.10▓▓░░░|2.78▓▓▓▓░|15.90▓▓▓▓▓░|1.78▓▓░░░░|18.67▓▓▓▓░░░░░░░░|0.24▓░░░░|
> |7|0.81▓▓░░░|3.19▓▓▓▓▓|13.25▓▓▓▓░░|3.24▓▓▓▓░░|20.22▓▓▓▓░░░░░░░░|0.67▓▓░░░|
> |9|0.57▓░░░░|3.19▓▓▓▓▓|18.55▓▓▓▓▓▓|4.75▓▓▓▓▓░|20.97▓▓▓▓░░░░░░░░|0.71▓▓░░░|
> |11|0.70▓▓░░░|2.86▓▓▓▓░|15.90▓▓▓▓▓░|5.19▓▓▓▓▓▓|59.81▓▓▓▓▓▓▓▓▓▓▓▓|0.91▓▓▓░░|
> |13|0.48▓░░░░|3.15▓▓▓▓▓|18.55▓▓▓▓▓▓|4.27▓▓▓▓▓░|26.71▓▓▓▓▓░░░░░░░|1.09▓▓▓▓░|
> |15|0.52▓░░░░|2.96▓▓▓▓▓|7.95▓▓▓░░░|1.35▓▓░░░░|18.96▓▓▓▓░░░░░░░░|1.28▓▓▓▓░|
> |17|0.58▓░░░░|2.81▓▓▓▓░|18.55▓▓▓▓▓▓|3.19▓▓▓▓░░|28.86▓▓▓▓▓▓░░░░░░|1.44▓▓▓▓▓|
> |19|0.47▓░░░░|0.00░░░░░|0.00░░░░░░|0.00░░░░░░|0.00░░░░░░░░░░░░|0.00░░░░░|
> |21|1.17▓▓▓░░|2.82▓▓▓▓░|5.30▓▓░░░░|5.10▓▓▓▓▓▓|14.52▓▓▓░░░░░░░░░|0.58▓▓░░░|
> |23|0.86▓▓░░░|0.00░░░░░|0.00░░░░░░|0.00░░░░░░|0.00░░░░░░░░░░░░|0.00░░░░░|
> |25|1.07▓▓░░░|2.97▓▓▓▓▓|2.65▓░░░░░|0.68▓░░░░░|1.79░░░░░░░░░░░░|1.43▓▓▓▓▓|
> |27|1.21▓▓▓░░|0.00░░░░░|0.00░░░░░░|0.00░░░░░░|0.00░░░░░░░░░░░░|0.00░░░░░|
> |29|0.68▓▓░░░|0.00░░░░░|0.00░░░░░░|0.00░░░░░░|0.00░░░░░░░░░░░░|0.00░░░░░|
> |31|1.16▓▓▓░░|3.07▓▓▓▓▓|10.60▓▓▓░░|2.10▓▓░░░░|23.30▓▓▓▓▓░░░░░░░|0.58▓▓░░░|
> |33|1.24▓▓▓░░|2.78▓▓▓▓░|2.65▓░░░░░|0.42░░░░░░|5.22▓░░░░░░░░░░░|0.56▓▓░░░|
> |35|1.45▓▓▓░░|0.00░░░░░|0.00░░░░░░|0.00░░░░░░|0.00░░░░░░░░░░░░|0.00░░░░░|
> |37|1.44▓▓▓░░|2.53▓▓▓▓░|2.65▓░░░░░|0.60▓░░░░░|10.31▓▓░░░░░░░░░░|0.55▓▓░░░|
> |39|0.76▓▓░░░|2.65▓▓▓▓░|2.65▓░░░░░|1.03▓░░░░░|4.14▓░░░░░░░░░░░|0.58▓▓░░░|
> |41|1.37▓▓▓░░|3.27▓▓▓▓▓|7.95▓▓▓░░░|0.39░░░░░░|8.50▓▓░░░░░░░░░░|0.56▓▓░░░|
> |43|2.04▓▓▓▓▓|2.84▓▓▓▓░|2.65▓░░░░░|0.34░░░░░░|1.38░░░░░░░░░░░░|0.57▓▓░░░|
> |45|1.08▓▓░░░|0.00░░░░░|0.00░░░░░░|0.00░░░░░░|0.00░░░░░░░░░░░░|0.00░░░░░|
> |47|1.97▓▓▓▓░|0.00░░░░░|0.00░░░░░░|0.00░░░░░░|0.00░░░░░░░░░░░░|0.00░░░░░|
> |49|2.21▓▓▓▓▓|0.00░░░░░|0.00░░░░░░|0.00░░░░░░|0.00░░░░░░░░░░░░|0.00░░░░░|
> |1-19|11.53|54.06|291.50|51.73|598.05|14.11|
> |1-150|217.51|190.47|651.90|145.49|1139.48|31.47|
>
> For a fair comparison, we report the wall-clock time required for both Flick and FedProx to reach the same target accuracy (i.e., best accuracy by FedProx): **0.48** hours for Flick (at round 19) and **1.24** hours for FedProx (at round 142). The accumulated additional computational overhead by Flick in the first 19 rounds is *0.28* hours, as shown at the bottom of the table. Moreover, after 150 rounds, Flick further improves the global model accuracy by 10.09% over FedProx, with only an additional *0.66* hours cost. These results highlight that the overhead introduced by Flick is well justified by its superior efficiency and accuracy improvements.
>
> For real-world deployment, we also provide the wall-clock time to the target accuracy when local training and summary running on an NVIDIA Jetson AGX Orin: it takes **0.96** hours for Flick while **2.37** hours for FedProx. The accumulated additional computational overhead is *0.62* hours, with the breakdown across the six components being *57.3s, 1220.0s, 291.5s, 51.7s, 598.1s, and 14.1s*, respectively. We emphasize the strong flexibility of Flick for inference overhead by allowing clients and the server to choose models according to their available resource budgets. Please see our further response [4/6] to the comment of Reviewer t7ek for the details.
>
> In terms of the API fees, we use the **gpt-4o-mini** model for local summary analysis. There are a total of 388 API requests across 150 rounds, at a cost of \\$0.05, with \\$0.03 for input tokens and \\$0.02 for output tokens.
>
> `[3/5] Why not fine-tune a global model on all synthetic data at the server? This would reduce communication to just a single round.`
>
> We clarify that simply fine-tuning a global model on synthetic data, particularly in a one-shot way, is a suboptimal solution for two main reasons: 1) obtaining a rich task-related synthetic dataset, especially within a single round, is privacy-sensitive and resource-intensive; and 2) the performance gain from global fine-tuning alone, as shown in Tables 1 and 3, is limited.
>
> In Flick, the fine-tuning dataset $\mathcal{G}_s$ is built from scratch and dynamically updated over rounds. In contrast, one-shot fine-tuning requires collecting a rich and substantial dataset within a single round, which leads to privacy risk and significant latency. For example, the baseline method FGL bootstraps all clients to caption all local images and then uploads them to the server-side generator to construct a large-scale fine-tuning dataset in the first round. As shown in Figure 14(a) (Appendix D), this results in ~0.6 hours of warm-up latency. Although the global model achieves high accuracy after the first round, its performance over time is suboptimal. Moreover, collecting large-scale local summaries in a single round raises privacy risks and relies on full client participation, which is often unrealistic in real-world settings.
>
> Moreover, global fine-tuning alone is insufficient. Flick introduces dual usage of synthetic data: the local data compensation progressively provides clients with additional samples to approach the IID condition, yields long-term benefits, while fine-tuning the global model offers a more straightforward yet slight improvement. As shown in Table 3, ablation of local data compensation causes a larger accuracy drop than removing global fine-tuning. Table 1 further supports this: Flick, which combines both local compensation and global fine-tuning, consistently outperforms FGL that is only based on global fine-tuning.
>
> `[4/5] Lack of experiments under IID settings, e.g. using datasets like CIFAR.`
>
> We clarify Flick is specifically designed for heterogeneous FL scenarios, which are far more realistic scenarios in life than ideal IID settings. In such heterogeneous settings, data silos are non-IID, exhibiting both label skew and domain shift. Such non-IID settings are practical and pose significant challenges to global model performance. In contrast, simple IID settings are idealized and offer limited space for improvement. Flick aims to bridge the gap between realistic non-IID distribution and the IID ideal cases through local data compensation. Besides, we emphasize that Flick explicitly alleviates the domain shift issue, which is not captured in datasets like CIFAR-10 or CIFAR-100. Therefore, evaluation on such datasets cannot reflect or validate the core capabilities and contributions of Flick.
>
> `[5/5] Questions and Suggestions.`
>
> We have provided in Appendix B.2 the details of the data generation budget $G$, LLMs, and image generators used for the two datasets in Table 1. Specifically, we set $G$=5, use **stable-diffusion-v1-5** for image generation, and employ **gpt-4o-mini** for generating text prompts. We appreciate the suggestion and will move this information from Appendix to the main text in future revisions, to make the paper easier to follow. The table below summarizes the number of samples across three datasets -- PACS (7,984 images, 20 clients), Office-Caltech (2,003 images, 8 clients), and DomainNet (12,740 images, 100 clients) -- used in our experiments. For each baseline, we report the number of generated and training images under Flick over 150 rounds (before `/`) and up to reaching the target accuracy (after `/`).
>
> ||PACS||Office-Cal.||DomainNet||
> |-|-|-|-|-|-|-|
> |Flick|#generated|#training|#generated|#training|#generated|#training|
> |w/FedAvg|1083`/`270|9344`/`8554|243`/`217|3008`/`2998|1127`/`292|16905`/`13080|
> |w/FedProx|1089`/`447|9384`/`8934|278`/`253|3233`/`3178|1133`/`407|16760`/`13305|
> |w/FedDyn|1166`/`413|9509`/`8554|252`/`192|3028`/`2983|1186`/`435|17110`/`13410|
> |w/FedNAR|1100`/`289|9330`/`8609|250`/`220|3318`/`3108|1299`/`492|16810`/`13845|

---

> ### Comment · Reviewer_28YM · 2025-08-01
>
> Thank you for your efforts. I have carefully read your response and would like to provide further comments as follows。
>
> Regarding weakness 1:
> 1. I understand that deploying a model like Qwen2.5-VL may require computation resources. However, in Flick, the client side also needs to host a VLP model, which likewise demands significant resources. Moreover, many recent MLLMs are relatively lightweight and can be deployed efficiently using frameworks such as vLLM or LMDeploy, which help keep resource requirements acceptable. Additionally, the field of knowledge distillation has seen extensive progress, enabling the compression of larger models into much smaller, more efficient versions.
> 2. The overall cost — including local training, image generation using diffusion models, prompt generation with LLMs, and global fine-tuning — incurs far greater overhead than simply using off-the-shelf LLMs. Since your method already relies on LLMs and LDMs, it would be more convincing to compare your approach with them in a few-shot setting. A more detailed comparison of the required resources and the resulting performance is necessary to support your claims.
>
> Therefore, I strongly suggest making comparisons with MLLMs like Qwen2-VL and Qwen2.5-VL.
>
> Regarding weakness 2:
> 1. Based on my experience with LLMs and diffusion models, the generation time for LDMs is typically much longer than 1 second per image. Could you clarify how you achieve this inference speed? The specific inference parameters and settings are missing from your paper — can you please provide these details?
> 2. More importantly, you have not provided any statistics on resource consumption, such as GPU usage. Given that Flick heavily relies on both LLMs and LDMs, its resource requirements are likely to be much higher than those of traditional federated learning. Additional analysis and comparisons are needed to properly justify the feasibility and efficiency of your approach.

---

> > ### Author Response · Authors · 2025-08-03
> > **Response [1/2]**
> >
> > Thank you for reviewing our rebuttal swiftly; we really appreciate it. Below are our responses.
> >
> > `[Preface] A general response to comments from the reviewer.`
> >
> > We are glad that responses [3/5] to [5/5] have addressed your concerns regarding the global model fine-tuning in Flick and have further clarified the heterogeneous data setting, hyperparameter choices, and synthetic data generation.
> >
> > We also understand your remaining concerns about the use of powerful generative models to enhance model performance in the FL setting. To address these concerns, we recognize it is important to clarify the following two questions:
> >
> > *Given the rapid advancement of versatile foundation models, what role do conventional models (e.g., ResNet) still play?*
> >
> > and the follow-up:
> >
> > *How can such powerful foundation models be leveraged to benefit the training of lightweight, task-specific models, especially in a privacy-preserving federation way?*
> >
> > By answering these two questions, we aim to highlight the contribution of our data-driven Flick in improving the performance under heterogeneous FL and to re-emphasize the practical value of leveraging generative models to assist task-specific model training, as previously discussed in response [1/5].
> >
> > First, while foundation models offer broad generalization capabilities across a wide range of tasks, directly deploying them in real-world applications, especially on resource-constrained edge devices, is still challenging (for general tasks) and is overkill for specific tasks. This underscores the demand for lightweight task-specific models that balance efficiency and effectiveness in deployment.
> >
> > However, the increasing availability of generative models brings great potential to improve the training of these lightweight models. In the FL context, conventional approaches focus on optimizing training over distributed data silos, which inherently limits the upper bound of accessible knowledge. This performance ceiling can be lifted by distilling external task-related knowledge embedded in foundation models, which is actually our target in Flick.
> >
> > In the following, we provide some experimental runtime results to support our claims and make the discussions more concrete.
> >
> > `[weakness 1] The overheads and resource requirements in Flick, especially compared to some LLMs.`
> >
> > **Q1: I understand that deploying a model like Qwen2.5-VL may require computation resources. However, in Flick, the client side also needs to host a VLP model, which likewise demands significant resources.**
> >
> > We clarify that the resource requirements for local image captioning in Flick are acceptable, primarily for two reasons. First, the VLP model used in Flick is significantly lighter than MLLM, even when compared with the lighter version Qwen2-VL-2B, both in terms of storage and runtime resource consumption (see the next paragraph for the comparison details). Second, and more importantly, each client in Flick only employs the VLP model to caption a limited number of local samples (i.e., one representative sample per class) during training. In contrast, the inference phase is performed only using a lightweight model (e.g., ResNet-18), which imposes far less computational overhead than using a large foundation model for inference (please refer to the second paragraph of the response to Q3 (below) for the comparison details).
> >
> > For quantitative evaluation, we compare the resource requirements of three MLLMs and three VLP models on the PACS dataset using an NVIDIA A40 GPU. For MLLMs performing image classification, **Qwen2.5-VL-7B** requires 16.57 GB of storage, 6.36 seconds of inference latency per sample, and 17.70 GB of GPU memory usage measured by the NVIDIA Management Library (*punvml*); **Qwen2.5-VL-3B** requires 7.51 GB of storage, 4.57 seconds per sample, and 8.08 GB of GPU memory; an a lighter **Qwen2-VL-2B** requires 4.41 GB of storage, 3.91 seconds per sample, and 5.62 GB of GPU memory.
> >
> > In contrast, the VLP models used for image captioning are much lighter: **BLIP-Large** requires 1.88 GB storage and 3.19 GB of memory usage; **BLIP-Base** requires 0.97 GB storage and 2.30 GB of GPU usage; and **ViT-GPT2** requires 0.96 GB storage and 2.48 GB of GPU memory usage. Moreover, Flick shows consistent performance gains across these three image captioning models. Such flexibility enables clients to choose models that are best suited to their available computational budgets.

---

> > > ### Author Response · Authors · 2025-08-03
> > > **Response [2/2]**
> > >
> > > **Q2: Many MLLMs are relatively lightweight and can be deployed using frameworks such as vLLM or LMDeploy, which help keep resource requirements acceptable. Additionally, knowledge distillation enables the compression of larger models into much smaller, more efficient versions.**
> > >
> > > We acknowledge that frameworks such as vLLM significantly reduce the latency and resource requirements for deploying LLM inference. Knowledge distillation (KD) methods also offer solutions for compressing larger models into smaller, more efficient ones with fewer parameters and lower computational costs. However, we argue that in practical deployment scenarios, even so-called lightweight MLLMs remain substantially more resource-intensive than traditional models such as ResNet-18, which typically occupies less than 50 MB. For instance, despite the efficiency gains from vLLM, models like Qwen2.5-VL cannot be deployed on highly resource-constrained embedded devices such as a Raspberry Pi. In contrast, ResNet-18 can be readily deployed on such devices, making them more suitable for real-world edge deployment.
> > >
> > > **Q3: Since your method already relies on LLMs and LDMs, it would be more convincing to compare your approach with them in a few-shot setting. Comparison of the required resources and the performance is necessary. Therefore, I strongly suggest making comparisons with MLLMs like Qwen2-VL and Qwen2.5-VL.**
> > >
> > > We do not question the strong performance of foundation models such as Qwen2-VL and Qwen2.5-VL on image classification tasks, including the PACS, Office-Caltech, and DomainNet benchmarks. However, their severe inference latency on resource-constrained devices makes them impractical for real-world deployment.
> > >
> > > To support our claims, we further conduct some more experiments, and below we report the inference latency on an NVIDIA Jetson AGX Orin. Under GPU-mode, **Qwen2.5-VL-3B** takes 7.05 seconds per sample, while the lighter **Qwen2-VL-2B** still requires 5.94 seconds. The latency is severely exacerbated under CPU-only mode: 239.43 seconds for the **Qwen2.5-VL-3B** and 169.93 seconds for the **Qwen2-VL-2B**. We emphasize that the Jetson AGX Orin is already considered powerful hardware for edge computing. In contrast, **ResNet-18**, used for inference in Flick, requires only 4.73 seconds per sample under the same CPU-only configuration. When using GPU acceleration, the inference time is further reduced to 0.4 seconds per sample.
> > >
> > > `[weakness 2] Detailed settings for LDMs in Flick and statistics on resource consumption.`
> > >
> > > **Q4: Could you clarify how you achieve this inference speed? The specific inference parameters and settings are missing from your paper.**
> > >
> > > In our experiments, we use the publicly available `sd-legacy/stable-diffusion-v1-5` model from Hugging Face. The image generation relies on `StableDiffusionPipeline` from the `diffusers` library. We set the default output resolution to 512$\times$512 pixels and the number of inference steps to 30, balancing quality and speed. The core data generation code is shown below:
> > >
> > > ```
> > > import torch
> > > from diffusers import StableDiffusionPipeline
> > >
> > > model_id = "sd-legacy/stable-diffusion-v1-5"
> > > pipe = StableDiffusionPipeline.from_pretrained(model_id, torch_dtype=torch.float16)
> > > pipe = pipe.to("cuda")
> > >
> > > images = pipe(text_prompt_list, num_inference_steps=30).images
> > > ```
> > >
> > > The main dependencies include `torch==2.7.1; torchvision==0.22.1; diffusers==0.34.0; accelerate==1.9.0`. On an NVIDIA A40 GPU, the average generation time is approximately **1.13 seconds per image**, with peak GPU memory usage of **5.69 GB**. We emphasize that data generation in the Flick is performed on the server side, which is typically endowed with powerful hardware.
> > >
> > > **Q5: You have not provided any statistics on resource consumption, such as GPU usage. Given that Flick heavily relies on both LLMs and LDMs, its resource requirements are likely to be much higher than those of traditional FL.**
> > >
> > > In response, we report that the peak GPU memory usage for the Stable Diffusion v1.5 (LDM) model on an NVIDIA A40 GPU is 5.69 GB. For the LLM, we are unable to directly measure GPU usage because we rely on the **gpt-4o-mini** model via the OpenAI API rather than running an LLM locally. Alternatively, we report the average response latency of **gpt-4o-mini** from prompt submission to reasoning and obtaining generated text prompts, which is approximately 2.6 seconds per request.
> > >
> > > Indeed, running LLM and LDM on the server introduces additional overhead. However, the time breakdown shown in response [2/5] indicates that such server-side overhead decreases over time, as Flick only 'heavily' requires data generation at the early edge. Furthermore, results in response [2/5] also show that Flick achieves both better time-to-accuracy performance than baseline methods and higher model accuracy in the long run. These results highlight that the overhead introduced by Flick is well justified by its superior efficiency and accuracy improvements.

---

> ### Comment · Reviewer_28YM · 2025-08-04
>
> Thank you for your detailed response. I have carefully read your replies, and I think you have provided very good explanations that address at least half of my concerns regarding the basic setup.
> Although I still hold my reservations, I am convinced that using LLMs/LDMs to train small models like ResNet can make sense in certain scenarios, as you pointed out—particularly in highly specific and resource-constrained cases.
>
> Therefore, I have raised my score from 2 to 3. And increase the sub scores for originality and significance.
>
> Additionally, regarding the comparison with few-shot settings using MLLMs such as Qwen: I understand that few-shot models incur more inference costs, but using an off-the-shelf MLLM requires no training at all. I still have concerns here—when I suggested this comparison, I actually meant comparing the performance. While efficiency is indeed difficult to compare (since Flick requires training whereas few-shot Qwen incurs higher inference costs), performance can and should be compared directly. That was the main point of my suggestion.
>
> Lastly, about my concern that “Given that Flick heavily relies on both LLMs and LDMs, its resource requirements are likely to be much higher than those of traditional federated learning”: in your Q5 response, you still did not provide an explicit resource usage comparison with traditional FL methods. I understand that Flick will inevitably require more resources, but they should remain within an acceptable range.
>
> I believe adding these comparisons will further justify the performance-efficiency trade-off of your work.
>
> Thank you again for your efforts in addressing my concerns. If the two comparisons mentioned above can be added, I will make sure to raise my score further in recognition of your efforts.

---

> > ### Author Response · Authors · 2025-08-06
> > **Further Response [1/2]**
> >
> > Thank you for reviewing our response. We sincerely appreciate your insightful feedback to enhance the quality of our work. To this end, we will incorporate the relevant discussion into the revised manuscript. Our further responses to the remaining questions are provided below.
> >
> > `[Response to Q3] While efficiency is indeed difficult to compare (since Flick requires training whereas few-shot Qwen incurs higher inference costs), performance can and should be compared directly.`
> >
> > We appreciate that the reviewer recognizes the practical value of lightweight, task-specific models such as ResNet, for which the training can benefit from leveraging powerful foundation models.
> >
> > In response, we evaluate the performance of four MLLMs (i.e., Qwen2-VL-7b, Qwen2-VL-2b, Qwen2.5-VL-7b, and Qwen2.5-VL-3b) on the three datasets used in our paper. The tables below report the test accuracy for each domain of the task, along with the average accuracy across all domains. For completeness, we also recap the performance of Flick and the baseline method FGL (shown at the bottom of each table), as both of them generate synthetic samples using LLMs on the server side.
> >
> > |||PACS||||
> > |:-|:-:|:-:|:-:|:-:|:-:|
> > |**Models**|Photo|Art Painting|Cartoon|Sketch|AVG|
> > |Qwen2-VL-7b|99.70|99.76|99.58|94.42|98.36|
> > |Qwen2-VL-2b|99.70|98.30|99.58|94.67|98.06|
> > |Qwen2.5-VL-7b|100|96.84|98.94|93.65|97.36|
> > |Qwen2.5-VL-3b|100|96.60|99.15|92.64|97.10|
> > |||||||
> > |FGL|88.96|87.69|97.62|85.09|89.84|
> > |Flick|91.99|94.59|97.74|93.91|94.49|
> >
> > |||Office-Caltech||||
> > |:-|:-:|:-:|:-:|:-:|:-:|
> > |**Models**|Amazon|Caltech|DSLR|Webcam|AVG|
> > |Qwen2-VL-7b|97.41|98.67|100|100|99.02|
> > |Qwen2-VL-2b|91.71|95.11|92.86|98.33 |94.50|
> > |Qwen2.5-VL-7b|97.93|99.56|100|100 |99.37|
> > |Qwen2.5-VL-3b|97.41|97.33|100|96.67 |97.85|
> > |||||||
> > |FGL|67.36|62.22|85.71|80.00|73.82|
> > |Flick|75.82|64.00|96.43|77.78|78.51|
> >
> > ||||DomainNet|||||
> > |:-|:-:|:-:|:-:|:-:|:-:|:-:|:-:|
> > |**Models**|Clipart|Infograph|Painting|Quickdraw|Real|Sketch|AVG|
> > |Qwen2-VL-7b|96.97|91.08|93.53|63.13|97.05|98.07|89.97|
> > |Qwen2-VL-2b|97.37|90.66|92.21|62.07|96.78|97.79|89.48|
> > |Qwen2.5-VL-7b|97.77|92.32|94.32|62.73|97.31|98.21|90.44|
> > |Qwen2.5-VL-3b|96.36|88.59|92.73|57.27|97.18|96.55|88.11|
> > |||||||||
> > |FGL|91.09|55.56|79.71|79.07|92.21|84.80|80.41|
> > |Flick|92.11|63.90|83.88|91.40|92.91|90.34|85.76|
> >
> > From these results, MLLMs indeed achieve strong performance, while Flick provides a global model with comparable inference accuracy on both the PACS and the more challenging DomainNet benchmarks. For the Office-Caltech dataset, we suspect the inferior performance of Flick is due to its highly imbalanced label distribution (i.e., a Dirichlet distribution with concentration parameter $\alpha$=0.05), and our choice of the task model, i.e., ResNet10, following prior works [1, 2].
> >
> > Moreover, we emphasize two points for fairly assessing the contribution of Flick: 1) training under heterogeneous FL settings inherently degrades model performance compared to centralized training; and 2) our data-driven Flick addresses such challenges more effectively than the similar LLM-based FGL and local generation approaches (Table 2 in the paper) by efficiently incorporating commonsense knowledge from LLMs.
> >
> > Finally, we note an interesting observation from the DomainNet results, particularly on the two challenging domains, 'Infograph' and 'Quickdraw': all four MLLMs perform relatively well on the 'Infograph' domain, while showing generally inferior performance on the 'Quickdraw' domain. As described in DomainNet [3], 'Quickdraw' images are collected from Google’s 'Quick, Draw!' game, where the drawings are very simple, abstract line sketches; 'Infograph' images are typically informational graphics such as posters, diagrams, and other structured visual layouts. This distinct performance gap between the two domains suggests that: 1) MLLMs inherently have a strong understanding of complex icons, symbols, and text arranged in a structured format; and 2) they still have limitations in handling certain tasks. This observation underscores the value of FL frameworks, which allow privacy-preserving model training for such domain-specific tasks using task-relevant, decentralized data.
> >
> > [1] Wenke Huang, et al. "Rethinking federated learning with domain shift: A prototype view". CVPR, 2023
> >
> > [2] Yuhang Chen, et al. "Fair federated learning under domain skew with local consistency and domain diversity". CVPR, 2024.
> >
> > [3] Xingchao Peng, et al. "Moment matching for multi-source domain adaptation". CVPR, 2019.

---

> ### Author Response · Authors · 2025-08-06
> **Further Response [2/2]**
>
> `[Response to Q5] An explicit resource usage comparison with traditional FL methods.`
>
> In response, we evaluate the resource usage from three aspects: 1) peak GPU memory (GB) during training, 2) GPU memory hours (GB·hour), akin to GPU hours but weighted by runtime memory usage, and 3) energy consumption (kilowatts·hour). We collect these statistics on an NVIDIA A40 GPU for both FedProx and Flick across the three benchmarks. The table below reports the per-round resource usage and the accumulated usage required to reach the target accuracy (i.e., best accuracy of FedProx). Since GPU memory hours and energy consumption vary across communication rounds, we provide both the dynamic range and the average value over all rounds.
>
> ||PACS||Office-Caltech||DomainNet||
> |:-|:-:|:-:|:-:|:-:|:-:|:-:|
> |**Metrics**|FedProx|+Flick|FedProx|+Flick|FedProx|+Flick|
> |Peak GPU memory (GB)|8.33|14.53|4.77|10.50|13.74|20.47|
> |GPU memory hour per round (GB·h)|[0.049, 0.064] Avg=0.056|[0.152, 0.353] Avg=0.223|[0.016, 0.022] Avg=0.020|[0.084, 0.376] Avg=0.101|[0.529, 0.681] Avg=0.637|[1.407, 2.110] Avg=1.887|
> |Energy per round (KWh)|[0.890, 80.62]$\times10^{-4}$ Avg=22.26$\times10^{-4}$|[5.641, 164.6]$\times10^{-4}$ Avg=68.30$\times10^{-4}$|[1.813, 52.47]$\times10^{-4}$ Avg=22.37$\times10^{-4}$|[1.894, 90.90]$\times10^{-4}$ Avg=27.56$\times10^{-4}$|[0.823, 27.36]$\times10^{-3}$ Avg=12.49$\times10^{-3}$|[0.806, 120.7]$\times10^{-3}$ Avg=47.44$\times10^{-3}$|
> |**When reaching the target accuracy**|||||||
> |Running time (h)|0.96|0.52|0.55|0.50|6.82|2.11|
> |GPU memory hour (GB·h)|7.91|5.92|2.63|5.25|91.79|34.61|
> |Energy (KWh)|0.149|0.075|0.063|0.050|1.329|0.444|
>
> For Flick, GPU memory hours and energy consumption show a wider dynamic range because the overhead from key components changes across rounds. As discussed in our previous response [2/5], Flick relies 'heavily' on LLM and LDM for synthetic data generation during the early training stage, which takes the majority of the overall cost. As training progresses, both GPU memory hours and energy consumption per round in Flick decrease. In contrast, the variation in these two per-round measurements for FedProx is mainly due to the randomness of participating clients in each round, where heterogeneous data silos introduce varying local training overhead.
>
> From the results, we observe that for the Office-Caltech dataset, Flick requires more GPU memory hours than FedProx to reach the target accuracy. This is because the additional GPU memory for LLM and LDM far exceeds that required for local training, thereby counteracting the resource savings from faster convergence (i.e., fewer communication rounds). However, Flick still offers shorter training latency to the target and achieves a higher best accuracy than FedProx. For more complex tasks, such as the DomainNet benchmark with large-scale clients and data, Flick outperforms conventional FL across all metrics: it reaches the same performance as counterpart FL frameworks with lower training latency, GPU memory usage, and energy cost; furthermore, it achieves better accuracy in the long run.
>
> To conclude, although Flick introduces additional overhead, primarily on the server, it enables faster convergence of the global model compared to FedProx. In most cases, Flick further reduces resource consumption to achieve the same target accuracy, particularly in challenging settings with large-scale clients and datasets. Furthermore, with continued training, Flick offers substantial accuracy gains; for example, on the PACS dataset, Flick achieves 11.43\% higher accuracy than FedProx after the same number of communication rounds.

---

> ### Comment · Reviewer_28YM · 2025-08-06
>
> Thank you for your hard work in addressing my concerns.
>
> The results you provided confirm with my expectations regarding Flick:
> 1. Compared to using off-the-shelf MLLMs such as Qwen, Flick yields lower performance. I suspect that with carefully selected examples, the performance gap could be even more pronounced.
> 2. Compared to traditional federated learning training using ResNet, Flick incurs much higher GPU usage.
>
> Despite these inherent limitations and my reservations about the practical value Flick may offer, I am convinced by the authors’ arguments that it presents a trade-off between performance and efficiency and may be valuable in certain scenarios.
>
> I also greatly appreciate the detailed experimental results that directly answer my questions. Therefore, I will raise my score to 4 to acknowledge the authors' efforts.
>
> ---
>
> I strongly recommend including these additional comparisons and statistics in the revised paper to better support and justify the positioning of Flick.
> Best of luck with your acceptance.

---

> ### Author Response · Authors · 2025-08-07
>
> Dear Reviewer 28YM,
>
> Thank you again for your time, great effort, and swift actions on reviewing our rebuttal and for the insightful discussions. Your constructive feedback has greatly helped us better reflect the novelty and contributions of our Flick. We will definitely add the suggested comparisons and resource usage statistics in the revised manuscript. Once again, we sincerely appreciate your detailed review, constructive suggestions, and timely engagement throughout the process.
>
> Best,
> Authors

---

### Official Review · Reviewer_t7ek · 2025-07-03

**Clarity:** 3
**Significance:** 3
**Originality:** 3
**Rating:** 5
**Confidence:** 3

**Summary:**

The paper introduces a novel framework aimed at addressing the challenges of data heterogeneity in Federated Learning (FL). Federated Learning allows for collaborative model training across decentralized data sources while preserving privacy. However, the heterogeneous nature of data distributions across different clients often leads to inconsistent learning objectives and suboptimal performance of the global model. Flick proposes a data-driven solution by incorporating commonsense knowledge from Large Language Models (LLMs) to generate synthetic data, which compensates for local data variations and enhances global model performance. The framework involves a local data summarization phase, where client-specific knowledge is captured in textual form, and a server-side data generation phase, where LLMs guide the generation of synthetic data used for fine-tuning the global model and aligning client data distributions. The paper presents extensive experimental results demonstrating improvements in model accuracy and convergence speed.

**Questions:**

Implementation Details:
Could you provide more details on the prompt generation process with LLMs? Specifically, how do you ensure that the prompts effectively capture the necessary client-specific knowledge without introducing significant privacy concerns? Additional technical details or examples would enhance understanding.
Real-World Applicability:
The reliance on synthetic data generation is a notable aspect of the framework. How do you envision Flick performing in real-world scenarios where data distributions may be more complex and nuanced? It would be useful to discuss potential strategies for adapting the framework to such environments.
Privacy Considerations:
While the paper outlines a low-sensitivity approach to data summarization, could you elaborate on any privacy assessments conducted? Understanding how the framework complies with privacy regulations and mitigates risks would be beneficial for its practical adoption.
Computational Resource Requirements:
Flick leverages server-side resources for data generation. How does the framework perform in environments with limited computational capabilities? Discussing any optimizations or adaptations for resource-constrained settings would be valuable.
The potential impact of synthetic data on model bias and fairness is not fully explored. Could you provide insights into how Flick addresses these issues, or suggest directions for future work to ensure equitable model performance across diverse client data?

**Ethical Concerns:**

["NO or VERY MINOR ethics concerns only"]

**Final Justification:**

I have checked out author’s responses and discussions, and decide to maintain my positive score.

**Limitations:**

The authors have addressed some limitations of their work, particularly regarding data heterogeneity and privacy considerations. However, further discussion on computational constraints and the implications of synthetic data on bias and fairness would strengthen the paper's impact and applicability.

**Quality:**

3

**Strengths And Weaknesses:**

Quality:
Strengths:
The methodology is robust and well-designed, integrating modern LLM capabilities with FL to address data heterogeneity.
Extensive experimental validation is conducted across multiple datasets, showcasing significant improvements in both accuracy and convergence speed.
The paper provides a comprehensive analysis of the framework's performance, including comparisons with existing methods and an ablation study.

Weaknesses:
The used benchmark dataset for evaluation may not fully capture the complexities of real-world FL deployments, where data distributions can vary unpredictably.
The implementation details of the LLM-guided prompt generation process could be expanded for greater clarity.

Clarity:
Strengths:
The paper is generally well-structured, with clear explanations of the Flick framework and its components.
Figures and tables effectively illustrate the problem, solution, and experimental results, aiding reader comprehension.

Weaknesses:
Certain technical aspects, such as privacy considerations and the specifics of text prompt generation, could be more thoroughly explained.
Some sections may benefit from additional background information to assist readers less familiar with the domain.

Significance:
Strengths:
The paper addresses a critical challenge in FL, offering a novel approach that could significantly enhance the applicability of FL in heterogeneous data environments.
The integration of LLMs for commonsense knowledge distillation introduces a new dimension to synthetic data generation, with potential applications beyond FL.

Weaknesses:
The framework's reliance on server-side resources may limit its applicability in scenarios with constrained computational infrastructure.
The potential impact of the generated synthetic data on model fairness and bias is not fully explored.

Originality:
Strengths:
The use of LLMs to guide synthetic data generation in FL is a novel approach that distinguishes Flick from existing methods.
The framework's design, which balances client-side and server-side responsibilities, offers a unique solution to data heterogeneity.

Weaknesses:
While innovative, the framework's dependence on LLMs could be seen as an extension of existing generative approaches, rather than a fundamentally new paradigm.

---

> ### Author Rebuttal · Authors · 2025-07-31
>
> We thank the reviewer for the insightful comments. Below are our responses.
>
> `[1/6] Implementation Details: Could you provide more details on the prompt generation process with LLMs? Specifically, how do you ensure that the prompts effectively capture the necessary client-specific knowledge without introducing significant privacy concerns?`
>
> In the data generation phase, Flick instructs the LLM to generate a set of class-specific text prompts using a predefined prompt template. These text prompts are then passed to a text-to-image model (e.g., LDM) to synthesize data. The prompt template, shown in Figure 3, takes as input the local summaries $\mathcal{T} ^{n}$, class $i$, and budget $G$ (number of text prompts). Appendix F also provides detailed examples of LLM responses for generating $G=5$ text prompts in the classes dog (Figure 15) and headphones (Figure 16), given the corresponding local captions.
>
> The privacy concerns mainly arise from the local summaries $\mathcal{T} ^{n}$, while these summaries are deliberately designed to contain low-sensitivity textual information. Extensive evaluation in Appendix E supports our privacy claims, and further discussion is provided in the response [3/6]. In a nutshell, $\mathcal{T} ^{n}$ is resistant to image reconstruction and satisfies an $\epsilon$-DP guarantee, making local summary a privacy-preserving design.
>
> `[2/6] Real-World Applicability: How do you envision Flick performing in real-world scenarios where data distributions may be more complex and nuanced?`
>
> We clarify that our framework design, particularly the calculation of decision matrix $\mathbf{D}$ (lines 200-207 in the paper), naturally enables its resilience to complex and heterogeneous real-world data distributions. Extensive experimental results support such a conclusion, including settings with both extreme label skew and domain shift, which are commonly used to simulate realistic non-IID scenarios.
>
> Specifically, data compensation for each local silo follows the matrix $\mathbf{D}$, where entries are obtained based on the performance of local models on a server-held, task-related validation dataset. Such a dataset is built from scratch and dynamically updated across communication rounds, making the data-driven compensation process fully automated and agnostic to the actual local distribution.
>
> To validate the performance of Flick under heterogeneous data distribution, we report experimental results under the Dirichlet distribution with the concentration parameter $\alpha=0.1$ for PACS dataset and $\alpha=0.05$ for Office-Caltech dataset, where a smaller $\alpha$ indicates stronger heterogeneity. We also evaluate Flick on the large-scale DomainNet dataset with 100 clients with $\alpha=0.1$ (Table 7 in the Appendix). Furthermore, we evaluate Flick under a more extreme case where each client holds samples from only two random classes (please see our response [4/4] to the comment of Reviewer Bcqt for the detailed results). Overall, Flick consistently improves convergence speed and accuracy of the global model by a large margin compared to the baseline methods, especially under more heterogeneous settings.
>
> `[3/6] Privacy Considerations: While the paper outlines a low-sensitivity approach to data summarization, could you elaborate on any privacy assessments conducted?`
>
> The local summary phase in Flick provides the server with essential client-specific information with minimal privacy risk by 1) selectively captioning only one representative sample per class and 2) using textual summaries that carry informative yet low-sensitivity content. We evaluate the privacy risk of such a data summarization approach from two perspectives: providing a formal $\epsilon$-DP guarantee and investigating the risk of raw data reconstruction from local summaries. As shown in Table 9 (Appendix E), we add Laplace noise (parameterized by $\epsilon$) to the textual summaries and observe that Flick maintains performance gain even under increasing noise levels. Moreover, Table 10 (Appendix E) shows image reconstruction results using a wide range of image generators. We observe that reconstructed images remain significantly dissimilar to the original raw data, as measured by three widely used image similarity metrics. It indicates that recovering the original data from local captions is highly challenging.
>
> `[4/6] Computational Resource Requirements: How does the framework perform in environments with limited computational capabilities?`
>
> We highlight two features of our Flick that support deployment in real-world scenarios: zero training burden and flexible inference overhead. Specifically, the four models within the framework, including the captioning VLP model, LLM, SBERT embedding model, and LDM model, are all out-of-the-box models without any fine-tuning or prompt-tuning when applied to the Flick framework. As a result, they introduce zero training burden to Flick. In terms of inference overhead, only the VLP model for image captioning runs on the edge side, while the other three models run on the server, which typically has rich computing resources. Moreover, Flick offers strong flexibility in resource requirements through: 1) tunable hyperparameters such as generation budget $G$ and server-held data pool size $\lvert \mathcal{G}_ s \lvert$, and 2) consistent performance gains across different choices of image captioning models, LLMs, and image generators, as shown in Figures 10, 11, and 12 in Appendix C, respectively. This flexibility enables both clients and the server to choose models that are best suited to their available computational budgets.
>
> `[5/6] The potential impact of synthetic data on model bias and fairness is not fully explored. Could you provide insights into how Flick addresses these issues, or suggest directions for future work to ensure equitable model performance across diverse client data?`
>
> The bias and fairness of synthetic data primarily depend on the text prompts fed into the text-to-image model. In Flick, the server uses a predefined prompt template to instruct LLM in generating these text prompts, as discussed in our response [1/6]. Within the template, the local summaries $\mathcal{T}^{n}$ capture the cross-client-specific knowledge. We emphasize that all participating clients have an equal opportunity to contribute to $\mathcal{T}^{n}$ in Flick, as each client shares the same number of local image captions with the server. Moreover, designing a fairness-aware prompt template could further mitigate the potential bias during text prompt generation and the subsequent data synthesis. We leave this as a promising direction for future work.
>
> `[6/6] While innovative, the framework's dependence on LLMs could be seen as an extension of existing generative approaches, rather than a fundamentally new paradigm.`
>
> While the availability of versatile generative models has made data synthesis more accessible, effective data generation in the context of FL remains non-trivial, primarily due to privacy and resource constraints. We highlight that Flick is a novel data-driven framework tailored to the challenges of heterogeneous FL. We solve the problem of designing a generative pipeline leveraging limited, low-sensitivity cross-client knowledge and task-relevant commonsense knowledge of LLMs.
>
> Rather than merely stringing together existing generative tools, Flick is carefully designed to address three core challenges: 1) distillation of client-specific knowledge in a low-sensitivity way; 2) prompt design that effectively guides the instillation of task-relevant knowledge from LLMs into the data generation process; 3) usage of the synthetic samples, balancing both effectiveness (long-term improvements in model performance) and efficiency (controllable and flexible overhead). The effectiveness of each key design module is thoroughly validated through the ablation study presented in Table 3.

---

> > ### Comment · Reviewer_t7ek · 2025-08-09
> >
> > Thanks to authors for the helpful responses. Authors have properly addressed my concerns and I decide to keep my positive scores.

---

> > > ### Author Response · Authors · 2025-08-09
> > >
> > > Dear Reviewer t7ek,
> > >
> > > Thank you for taking the time to review our rebuttal. We are glad that our responses have addressed your concerns.
> > >
> > > Best,
> > > Authors

---

### Official Review · Reviewer_yj5i · 2025-07-03

**Clarity:** 3
**Significance:** 3
**Originality:** 3
**Rating:** 4
**Confidence:** 3

**Summary:**

The paper proposes Flick, a data-driven framework designed to address data heterogeneity across silos in federated learning. Flick leverages commonsense knowledge from large language models (LLMs) to generate synthetic data, enhancing data diversity and balance. To protect local privacy, clients share only textual captions describing their local data with the server, rather than the raw data itself. The server then uses these captions to create text prompts via an off-the-shelf LLM, which guides a generative model to produce synthetic data for both server-side augmentation and local use.

**Questions:**

- The retrieval mechanism seems to introduce duplicate generated samples across different clients or even on the same client. Could you provide quantitative insights on how frequently this duplication occurs in practice? Additionally, how does this duplication impact model performance, either positively (e.g., through regularization) or negatively (e.g., by reducing data diversity)?
- The paper mentions that stale sample removal is applied only to the server-side dataset. Would applying a similar strategy to the client-side local datasets also be beneficial? Could you clarify the rationale for restricting this operation to the server dataset, and whether client-side stale samples could similarly degrade learning performance?
- In line 126, the paper suggests that client datasets progressively approximate an IID distribution over time. However, this assumption may not hold if certain classes inherently present greater learning difficulty. Could you expand on how the proposed method addresses or accounts for such class-specific learning challenges in the non-IID setting?
- In line 166, the sample selection strategy favors choosing samples with the smallest loss for underperforming classes, under the assumption that such samples are less likely to be noisy. This makes sense in early training stages. However, in later stages, prioritizing harder (i.e., higher-loss) samples might better facilitate learning. Could you clarify why your method consistently favors small-loss samples?

**Ethical Concerns:**

["NO or VERY MINOR ethics concerns only"]

**Final Justification:**

Authors have properly addressed my concerns. I would like to maintain my positive score.

**Limitations:**

yes

**Quality:**

4

**Strengths And Weaknesses:**

Strengths
- The paper is well-written and easy to follow.
- The evaluation is thorough, providing comprehensive results that help assess the effectiveness of the proposed approach.

Weaknesses
- Since federated learning often involves private or domain-specific datasets, off-the-shelf LLMs may lack the relevant commonsense knowledge to accurately represent or augment such data.
- The motivation behind certain design choices and methodological components is not clearly explained. (See the detailed questions below for clarification.)

---

> ### Author Rebuttal · Authors · 2025-07-31
>
> We thank the reviewer for the insightful comments. Below are our responses.
>
> `[1/5] Since federated learning often involves private or domain-specific datasets, off-the-shelf LLMs may lack the relevant commonsense knowledge to accurately represent or augment such data.`
>
> Typically, the off-the-shelf LLM (e.g., gpt-4o-mini used in our experiments) contains rich knowledge spanning a wide range of domains. We want to highlight that our proposed pipeline in Flick can effectively query task-relevant information from such an LLM and instill it into the data generation process. Furthermore, in practical deployments, the server in Flick can substitute the off-the-shelf LLM with a task-oriented LLM to better capture domain-relevant commonsense knowledge.
>
> `[2/5] The retrieval mechanism seems to introduce duplicate generated samples across different clients or even on the same client. Could you provide quantitative insights on how frequently this duplication occurs in practice? Additionally, how does this duplication impact model performance, either positively (e.g., through regularization) or negatively (e.g., by reducing data diversity)?`
>
> There indeed exist some duplicate generated samples that can occur both within a client and across different silos. We define the intra-client duplication as the proportion of duplicated samples within a single client, and inter-client duplication as the proportion of duplicated samples shared across clients. For example, in our PACS experiment using FedAvg as the baseline (Table 1), the intra-client duplication across all clients ranges from 0% to 2.86% (with an average of 1.35%), while the inter-client duplication ranges from 4.89% to 19.30% (with an average of 12.79%).
>
> While such duplication may slightly reduce data diversity, it also brings benefits by reducing generation overhead through sample reuse. This reflects a trade-off between the efficiency and effectiveness of the data generation process. The degree of duplication is controlled by the hyperparameter $T_ s$, which defines the text similarity threshold for data retrieval. As shown in Figure 9(a), a higher $T_ s$ reduces duplication by favoring the generation of new samples over retrieval. We observe consistent performance gain by Flick across $T_ s$ ranging from 0.6 to 0.9, and we set $T_ s=0.8$ as the default in our experiment.
>
> `[3/5] The paper mentions that stale sample removal is applied only to the server-side dataset. Would applying a similar strategy to the client-side local datasets also be beneficial? Could you clarify the rationale for restricting this operation to the server dataset, and whether client-side stale samples could similarly degrade learning performance?`
>
> We clarify that the update mechanism for the server-side data pool $\mathcal{G}_s$ is tightly tied to its function. Removing stale samples helps prevent overfitting problems and ensures that the dynamic $\mathcal{G}_s$ remains effective both as a validation set for evaluating local updates and a fine-tuning dataset for enhancing the global model. Moreover, this strategy keeps the retrieval and storage overhead constant, as the size of $\mathcal{G}_s$ remains stable across communication rounds.
>
> In contrast, applying such an update mechanism to the client-side local dataset $\mathcal{D}_j$ is not suitable, since $\mathcal{D}_j$ serves as the training dataset where every sample matters. After local compensation in Flick, we expect the local silos to become domain-rich and progressively closer to an IID distribution, making stale sample removal on the client side unnecessary.
>
> `[4/5] In line 126, the paper suggests that client datasets progressively approximate an IID distribution over time. However, this assumption may not hold if certain classes inherently present greater learning difficulty. Could you expand on how the proposed method addresses or accounts for such class-specific learning challenges in the non-IID setting?`
>
> We clarify that the client in Flick progressively approximate an IID distribution through local compensation, which holds even in the presence of “hard” classes. Specifically, Flick tends to compensate for such hard classes across all participating clients via the decision matrix $\mathbf{D}$ (lines 202-207), since such learning challenges are not client-specific but lead to generally underperforming local models on validation samples of these classes. In such cases, the local data still approximates an IID distribution. We want to highlight the distinction between an IID and a balanced distribution: Flick compensates the local data across clients towards an identical distribution (i.e., $\mathcal{P}_ {j_ 1}(y)\simeq$ $\mathcal{P}_ {j_ 2}(y)$ with $ \forall j_ 1, j_ 2 \in \mathcal{J}, j_ 1 \neq j_ 2$), while the resulting label distribution ($\mathcal{P}_ {j_ 1}(y)$ and $\mathcal{P}_ {j_ 2}(y)$) may still be imbalanced.
>
> `[5/5] In line 166, the sample selection strategy favors choosing samples with the smallest loss for underperforming classes, under the assumption that such samples are less likely to be noisy. This makes sense in early training stages. However, in later stages, prioritizing harder (i.e., higher-loss) samples might better facilitate learning. Could you clarify why your method consistently favors small-loss samples?`
>
> We clarify that our strategy does not consistently favor small-loss samples. Specifically, for well-performing classes, we intentionally select the sample with the largest training loss for image captioning, as such samples provide larger gradients and thus contribute more to model updates. Moreover, the choice between small-loss and large-loss samples varies across communication rounds, even for the same class within the same client, depending on the dynamic relationship between the class-wise loss and the averaged training loss (lines 161-166).

---

> > ### Comment · Reviewer_yj5i · 2025-08-06
> >
> > Thank you for your detailed response. Most of my concerns have been addressed, but I still have a few remaining points.
> >
> > Regarding the last point about the sample selection strategy, I’d like to clarify my question further. In the later stages of training, do you think it would be more effective to prioritize harder samples (those with the highest loss) from both underperforming and above-average classes? My reasoning is that, at later stages, the informativeness of samples with low loss may be minimal compared to earlier stages.
> >
> > As for my first question, I still believe the proposed method may face limitations in certain scenarios—for example, when a private company wants to train solely on their own domain-specific data (e.g., detecting defects in proprietary products). Nonetheless, I agree that the method is well-suited for many general federated learning scenarios.
> >
> > Overall, I remain positive about the paper.

---

> > > ### Author Response · Authors · 2025-08-07
> > > **Response to comments**
> > >
> > > Thank you for reviewing our rebuttal. We are glad that responses have addressed your concerns regarding data retrieval, data pool updates, and local data compensation in Flick. Below are our responses and further clarification.
> > >
> > > `Q1: In the later stages of training, do you think it would be more effective to prioritize harder samples (those with the highest loss) from both underperforming and above-average classes? My reasoning is that, at later stages, the informativeness of samples with low loss may be minimal compared to earlier stages.`
> > >
> > > Thank you for suggesting this insightful sample selection strategy in Flick. Below, we discuss the feasibility and effectiveness of prioritizing harder samples (i.e., those with higher loss) during the later stages of training.
> > >
> > > Feasibility: Implementing this strategy requires recognizing when the training process has reached its later stages. A common heuristic is to monitor the performance of the global model, e.g., accuracy on a validation set, as an indicator. However, in practical FL settings, maintaining a representative and high-quality validation set on the server side is non-trivial. Moreover, even when such a validation set exists, its quality and size directly affect the reliability of the performance estimates, making it challenging to precisely trigger stage-specific strategies.
> > >
> > > Effectiveness: Supposing the global model's performance is available and is used to trigger the strategy once a predefined target accuracy is reached, we conduct additional experiments on the PACS dataset. We observe marginal differences between the two strategies: the original strategy used in the paper offers 94.49\% of the best global model accuracy with a total number of 1083 generated samples, while the new strategy offers an accuracy of 93.87\% with 1135 generated samples. We attribute this to the fact that Flick 'heavily' generates synthetic samples at the early stage. When the global model reaches the target accuracy, the local summary, by captioning the selected samples, imposes minimal impact on the synthetic data generation.
> > >
> > > To conclude, while the ablation study in Table 6 (Appendix C.2) shows that a loss-based selection strategy outperforms random selection by improving global model performance and reducing generation overhead, further design on this strategy yields only limited differences. The original strategy used in the paper offers both its feasibility and effectiveness in selecting representative samples.
> > >
> > > `Q2: I still believe the proposed method may face limitations in certain scenarios—for example, when a private company wants to train solely on their own domain-specific data (e.g., detecting defects in proprietary products).`
> > >
> > > Thank you for highlighting this valuable use case. We agree that in certain scenarios, such as when a service provider possesses its own domain-specific dataset, centralized training on such data may seem preferable. However, it largely depends on the scale and quality of the available dataset. On one hand, while centralized training is possible, collecting a rich and high-quality dataset is often labor-intensive and costly, especially when manual labeling by expert annotators or paid workers is required. On the other hand, in many real-world situations, the data owned by the service provider may be insufficient to support the training of well-performing models. It necessitates privacy-preserving training across distributed data sources, thereby leading to the emergence of FL. Nevertheless, the model's performance is inherently degraded by the heterogeneous data silos. Our Flick addresses these challenges by generating synthetic data, instilling both client-specific knowledge and task-relevant commonsense knowledge from LLMs.
> > >
> > > As you correctly point out, general-purpose LLMs may lack the domain-specific knowledge required for specialized applications. To address this, Flick allows the server to substitute the general-purpose LLM with a domain-specific one. We recognize, however, that suitable domain-specific LLMs may not be readily available for varying tasks. We will include this discussion in the revised paper to clarify that Flick is well-suited for general-purpose scenarios, while also offering the flexibility to be adapted to other domain-specific applications.

---

### Official Review · Reviewer_Bcqt · 2025-07-14

**Clarity:** 2
**Significance:** 2
**Originality:** 2
**Rating:** 4
**Confidence:** 3

**Summary:**

The authors propose a novel framework named Flick, designed to address the heterogeneity of classic data silos in federated learning. It leverages the common-sense knowledge of LLMs to guide data generation, thereby resolving label and feature distribution shifts. Specifically, clients perform local data summarization to capture client-specific knowledge in textual form. The server then extracts high-quality, task-relevant knowledge from the LLM and uses it to generate informative textual prompts. These prompts guide the generative model in producing synthetic data, enabling global model fine-tuning and local data compensation.

**Questions:**

1. How can the generated summaries be evaluated?
2. Are there more reasonable selection methods for sample choice?
3. During sample annotation, the VLP model suffers from semantic loss when converting images to text. Could this lead to missing key features in the textual summaries?

**Ethical Concerns:**

["NO or VERY MINOR ethics concerns only"]

**Final Justification:**

Thanks for authors' rebuttal. most of my concerns have been addressed. I keep my score.

**Limitations:**

yes, but it's given in supplementary material. it should be mentioned briefly in main text.

**Paper Formatting Concerns:**

Null

**Quality:**

2

**Strengths And Weaknesses:**

Strengths：
1. The experimental results show improvement, demonstrating the method's effectiveness.
2. The combination of LLMs and federated learning (FL) is novel, with a reasonable and effective approach.
3. Enhanced privacy: clients only share text summaries, preventing the leakage of raw data.
4. The study includes comparisons with baseline models in terms of accuracy and convergence speed, as well as results in heterogeneous scenarios. Ablation experiments and a Performance across local generative methods comparison are also conducted.

Weaknesses：
1. The use of local summarization methods may lead to the loss of key features and insufficient representation of heterogeneity, thereby introducing new issues during the generation process.
2. During sample selection, there is a risk of failing to capture intra-class diversity, as well as overfitting or selecting low-quality data.
3. More evidence is needed to support privacy claims, and experiments under extreme data heterogeneity are lacking.

---

> ### Author Rebuttal · Authors · 2025-07-31
>
> We thank the reviewer for the insightful comments. Below are our responses.
>
> `[1/4] The use of local summarization methods, i.e., sample annotation by VLP model, may lead to the loss of key features and insufficient representation of heterogeneity, thereby introducing new issues during the generation process. How can the generated summaries be evaluated?`
>
> When generating the local summaries, we aim at an underlying trade-off among informativeness, privacy, and overhead. Specifically, we aim to ensure that the textual summaries are low sensitive, thereby making it difficult for the server to reconstruct the original data (as discussed in Table 10 in Appendix E). Besides, the overhead of image captioning constrains the scale of local samples for summarization. Therefore, in our design, each client follows a loss-based rule to selectively caption one representative data point per class, thus providing the server with effective yet privacy-preserving local summaries at an acceptable cost. The effectiveness of the local summary is also resilient to the choice of the VLP model, as demonstrated by the results in Figure 10 in Appendix C.
>
> While designing a reasonable metric to directly and quantitatively evaluate the quality of the summaries is non-trivial, the ablation results in Table 3 (w/o local summary) and Table 6 (random local summary) show a clear accuracy drop, which strongly shows the effectiveness of our local summarization design.
>
> `[2/4] During sample selection, there is a risk of failing to capture intra-class diversity, as well as overfitting or selecting low-quality data. Are there more reasonable selection methods for sample choice?`
>
> On one hand, each participating client selects one representative data point per class with minimal captioning overhead. We clarify that the selected sample varies across the communication rounds, even for the same class within the same client, which benefits capturing intra-class diversity in the long run. On the other hand, the loss-based selection strategy ensures that clients choose clean, undistorted samples for underperforming classes and impactful samples for well-performing classes, thereby avoiding captioning low-quality data. The ablation results shown in Table 6, where random sample selection leads to a clearly lower accuracy, further validate the effectiveness of our design.
>
> While methods such as hard-sample selection [1] could potentially improve the sample selection in Flick, directly adopting them is not suitable due to the overhead constraints and the limited, imbalanced nature of data silos in our case. We leave this as a promising direction for future work.
>
> [1] Angelos Katharopoulos and François Fleuret. "Not all samples are created equal: Deep learning with importance sampling", ICML, 2018.
>
> `[3/4] More evidence is needed to support privacy claims.`
>
> We have already investigated the privacy guarantee of our selective sample captioning mechanism from two perspectives: 1) the formal $\epsilon$-DP guarantee (Table 9 in Appendix E), and 2) the potential risk of reconstructing raw data from the textual captions (Table 10 in Appendix E). We would appreciate it if the reviewer could clarify the additional evidence that should be further considered to support our privacy claims.
>
> `[4/4] Experiments under extreme data heterogeneity are lacking.`
>
> In response, we conduct additional experiments under a more extreme heterogeneous setting, where each client holds samples from only two random classes on the large-scale DomainNet dataset with 100 clients. We report the results in the following table.
>
> |Methods|Clipart|Infograph|Painting|Quickdraw|Real|Sketch||AVG|Δ↑|#Round↓|
> |-|-|-|-|-|-|-|-|-|-|-|
> |*FedAvg*|70.11|45.02|62.00|64.22|84.19|63.86||65.03±0.20|–|143|
> |+FedBN|71.28|42.11|61.22|65.93|83.26|61.59||64.23±0.46|-0.08|-|
> |+FedHEAL|70.24|41.29|62.48|75.40|81.55|62.34||65.55±0.32|0.52|84|
> |+DynaFed|71.93|42.71|67.49|70.23|82.93|61.41||66.28±0.27|1.25|95|
> |+FedFTG|71.99|45.95|62.73|65.87|81.09|63.31||65.15±0.40|0.12|90|
> |+FGL|76.52|49.72|71.86|57.98|90.98|74.16||70.20±0.79|5.17|19|
> |**+Flick**|84.01|53.63|76.68|81.73|90.12|82.34||**78.09±0.12**|**13.06**|**18**|
>
> Compared to the Dirichlet distribution with $\alpha=0.1$ (Table 8 in Appendix C.4), we can observe that the accuracy of FedAvg drops from 74.13% to 65.03%, indicating an extreme heterogeneity under this setup. The experimental results show that our Flick consistently outperforms the baseline methods, and the performance gain is actually more significant as the data distribution across clients becomes extremely heterogeneous.

---

> > ### Comment · Reviewer_Bcqt · 2025-08-07
> > **Response to authors' rebuttal**
> >
> > Thanks for authors' rebuttal. most of my concerns have been addressed. I keep my score.

---

> > > ### Author Response · Authors · 2025-08-07
> > >
> > > Dear Reviewer Bcqt,
> > >
> > > Thank you for taking the time to review our rebuttal. We are glad that our responses have addressed your concerns. We would be happy to provide any additional follow-up clarifications on questions you may have.
> > >
> > > Best,
> > > Authors

---

### Decision · Program_Chairs · 2025-09-17

**Decision:**

Accept (poster)

**Comment:**

# Summary
The paper introduces Flick, a novel framework for federated learning addressing data heterogeneity through synthetic data generation guided by commonsense knowledge of Large Language Models (LLMs). Clients summarize local data to capture client-specific knowledge in textual form, which is used by the server to create informative prompts for generating synthetic data. This approach enables global model fine-tuning and local data compensation, enhancing model performance and convergence speed across diverse data sources in federated learning. Experiments show that Flick improves the global model accuracy by up to 11.43%, and accelerates convergence by up to 12.9x.

# Strengths
-  LLMs-based commonsense knowledge distillation introduces a novel dimension to synthetic data generation for both the global server and local clients in a federated learning setting.
- Clients share only text summaries, enhancing privacy protection by preventing the leakage of raw data.
- Extensive experiments with nine baseline models on two datasets provide a robust evaluation.
- The paper is well-written and easy to follow, with a clear structure and explanations of the Flick framework and its components.

# Weaknesses
- Local clients' textual summarization of local data might remove important details and information.
- The motivation behind certain design choices and methodological components is not clearly explained.
- More analysis of the data selection method in terms of data diversity and quality is needed. It is also important to discuss and compare different data selection criteria and their effects on different training stages.
- It is not clear if the commonsense knowledge of LLMs can cover sufficient domain-specific expert knowledge required in many heterogeneous settings for different clients.
- Several important design choices and technical details are not discussed and justified thoroughly.
- The focused image classification tasks might be too simple given the complexity and generalizability of LLMs and diffusion models.
- Computational overhead caused by the proposed method is not analyzed.
- Lack of comparison with some important baselines.

# Reasons to Accept
- The paper studies a novel application of LLMs to address the data synthesis in a heterogeneous federated learning setting.
- The proposed method leverages LLMs' textual summarization and prompting capabilities to reduce the privacy leakage risk.
- Comparisons with baselines and ablation studies are comprehensive for the focused image classification tasks.

# Discussion Summary
- In the rebuttal, the authors provided various new experimental results to answer the questions from the reviewers and further clarified several key points and details of the proposed method.
- In the responses by reviewers, though some concerns/questions remain, most reviewers confirmed that important concerns have been well addressed by the rebuttal, making them remain positive about the paper.